# PRINCIPLED RL FOR DIFFUSION LLMS EMERGES FROM A SEQUENCE-LEVEL PERSPECTIVE

**Jingyang Ou**[1,2,3], **Jiaqi Han**[4], **Minkai Xu**[4], **Shaoxuan Xu**[1,2,3],
**Jianwen Xie**[5], **Stefano Ermon**[4], **Yi Wu**[6†], **Chongxuan Li**[1,2,3†]

[1] Gaoling School of Artificial Intelligence, Renmin University of China
[2] Beijing Key Laboratory of Research on Large Models and Intelligent Governance
[3] Engineering Research Center of Next-Generation Intelligent Search and Recommendation, MOE
[4] Stanford University [5] Lambda, Inc [6] Tsinghua University † Corresponding authors
{oujingyang,chongxuanli}@ruc.edu.cn; jxwuyi@mail.tsinghua.edu.cn

## ABSTRACT

Reinforcement Learning (RL) has proven highly effective for autoregressive language models, but adapting these methods to diffusion large language models (dLLMs) presents fundamental challenges. The core difficulty lies in likelihood approximation: while autoregressive models naturally provide token-level conditional probabilities essential for token-level RL objectives (e.g., GRPO), dLLMs generate sequences through iterative non-autoregressive denoising steps that lack this factorization. To address this fundamental mismatch, we propose ELBO-based Sequence-level Policy Optimization (ESPO), a principled RL framework that treats entire sequence generation as a single action and uses the ELBO as a tractable sequence-level likelihood proxy. Our method incorporates per-token normalization of importance ratios and robust KL-divergence estimation to ensure stable large-scale training. Extensive experiments on mathematical reasoning, coding, and planning tasks demonstrate that ESPO significantly outperforms token-level baselines, achieving dramatic improvements of 20-40 points on the Countdown task, while maintaining consistent gains on math and coding benchmarks. Our approach establishes sequence-level optimization as a principled and empirically effective paradigm for RL in dLLMs. Our code is available at https://github.com/ML-GSAI/ESPO.

## 1 INTRODUCTION

Large language models (LLMs) (OpenAI, 2023) have become a cornerstone of modern natural language processing, achieving remarkable progress across math (Guo et al., 2025), coding (Hui et al., 2024), and planning tasks (Yao et al., 2023). While autoregressive (AR) modeling has long dominated this field, recent advances in *diffusion large language models* (dLLMs) have demonstrated strong potential as an alternative formulation (Ou et al., 2024; Shi et al., 2024; Sahoo et al., 2024; Nie et al., 2025; Ye et al., 2025). By modeling language generation as an iterative denoising process, dLLMs bypass the left-to-right dependency of AR models and offer advantages in long context (Liu et al., 2025b), multimodal (Yang et al., 2025; You et al., 2025a; Li et al., 2025; Yu et al., 2025) and fast inference (Inception Labs et al., 2025; DeepMind, 2025; Song et al., 2025).

With the advent of powerful pretrained dLLMs, the next frontier lies in post-training (Ouyang et al., 2022) to further enhance their capabilities. Among various post-training paradigms, reinforcement learning (RL) has emerged as a powerful approach that enables test-time scaling (Snell et al., 2025) through verifiable rewards (Guo et al., 2025). It has yielded substantial gains on reasoning tasks in recent AR models (OpenAI, 2024), such as math (Cobbe et al., 2021b), coding (Chen et al., 2021), and reasoning (Liu et al., 2023b). Motivated by this success, a natural question arises: how can we extend reinforcement learning to dLLMs?

Applying RL to dLLMs, however, is nontrivial. Mainstream RL algorithms in language modeling (e.g., GRPO (Shao et al., 2024)) assume a left-to-right factorization of the sequence likelihood

and rely on token-level importance ratios $\frac{\pi_\theta(y^k|x,y^{<k})}{\pi_{\theta_{\text{old}}}(y^k|x,y^{<k})}$. In contrast, dLLMs generate sequences non-autoregressively, making such token-level conditionals either ill-defined or computationally expensive. Prior attempts to address this discrepancy have resorted to heuristic approximations—such as mean-field surrogates (Zhao et al., 2025) or token-level ELBO contributions (Yang et al., 2025; Gong et al., 2025)—or else computationally heavy trajectory-level formulations (Huang et al., 2025). None of these approaches fully resolves the mismatch between autoregressive RL objectives and the holistic generation process of dLLMs.

In this work, we address this fundamental conflict by introducing *ELBO-based Sequence-level Policy Optimization (ESPO)*, a sequence-level reinforcement learning framework tailored for dLLMs. Our key insight is that dLLMs should not be forced into an autoregressive token-level action space. Instead, we treat the generation of an entire sequence as a single action, leveraging the ELBO as a tractable proxy for the intractable sequence log-likelihood. We further incorporate stabilization techniques essential for large-scale training, including per-token normalization of the ELBO ratio and robust KL-regularization. Our method eliminates the inconsistencies introduced by heuristic token-level approximations and enables stable, computationally efficient training.

Empirically, we validate the effectiveness of our design through extensive ablation studies, which confirm that combining sequence-level optimization with the ELBO objective provides a stable and principled foundation for reinforcement learning in dLLMs. Beyond ablations, we further evaluate our method on mainstream tasks spanning mathematics, coding, and planning. Across both LLaDA (Nie et al., 2025) and Dream (Ye et al., 2025), our approach consistently outperforms prior dLLM-RL baselines such as d1 (Zhao et al., 2025) and wd1 (Tang et al., 2025), with particularly strong gains on planning tasks that require global consistency.

In summary, we make the following contributions:

- We provide a systematic analysis of why standard autoregressive RL objectives are incompatible with the non-autoregressive dLLMs, clarifying the limitations of existing heuristic approaches.
- We propose ESPO, a principled sequence-level RL framework that leverages the ELBO as a tractable proxy for sequence likelihood and introduces stabilized ratio and KL estimators for robust large-scale training.
- We demonstrate through comprehensive experiments and ablation studies that ESPO yields stable optimization and consistent improvements across math, coding, and planning benchmarks, surpassing prior dLLM-RL methods.

## 2 BACKGROUND

### 2.1 DIFFUSION LARGE LANGUAGE MODELS

Diffusion Large Language Models (dLLMs), or more specifically Masked Diffusion Models (MDMs), define a forward process that gradually corrupts the clean input by replacing tokens with the mask token M. Given the prompt $x$ and the clean completions $y$, the forward process $q_t(y_t \mid y, x)$ at time $t$ is defined as follows:

$$q_t(y_t|y,x) = \prod_{i=1}^{L} q_t(y_t^i|y^i,x) \quad \text{and} \quad q_t(y_t^i|y^i,x) = \begin{cases} 1-t, & y_t^i = y^i, \\ t, & y_t^i = \text{M}. \end{cases} \tag{1}$$

Unlike autoregressive models, the exact log-likelihood $\log \pi_\theta(y|x)$ in dLLMs is typically approximated via the evidence lower bound (ELBO) (Ou et al., 2024; Shi et al., 2024; Sahoo et al., 2024):

$$\mathcal{L}_\theta(y|x) \triangleq \mathbb{E}_{t\sim\mathcal{U}[0,1]}\mathbb{E}_{y_t\sim q_t(y_t|y,x)}\left[\frac{1}{t}\sum_{i=1}^{L}\mathbf{1}[y_t^i = \text{M}]\log p_\theta(y^i|y_t,x)\right] \le \log \pi_\theta(y|x). \tag{2}$$

As noted in Ou et al. (2024); Nie et al. (2025); Zhu et al. (2025), Eq. (2) has an equivalent, lower-variance variant that replaces the continuous masking ratio $t$ with a discrete number of masked tokens $l$:

$$\mathcal{L}'_\theta(y|x) \triangleq \mathbb{E}_{l\sim\mathcal{U}(\{1,2,...,L\})}\mathbb{E}_{y_l\sim q_l(y_l|y,x)}\left[\frac{L}{l}\sum_{i=1}^{L}\mathbf{1}[y_l^i = \text{M}]\log p_\theta(y^i|y_l,x)\right], \tag{3}$$

where $l$ denotes the number of tokens masked (sampled uniformly), and $y_l$ represents the corrupted sequence obtained by masking $l$ tokens without replacement. While Eq. (2) and Eq. (3) only provide a bound, the ELBO has been empirically shown to be a tight and practically effective surrogate for the intractable exact likelihood. This holds true for both likelihood-based evaluation (e.g., perplexity (Lou et al., 2024)) and training (e.g., DPO variants (Zhu et al., 2025)).

## 2.2 REINFORCEMENT LEARNING

Policy gradient methods have been shown to be highly effective for post-training LLMs (Ouyang et al., 2022). Among them, Group Relative Policy Optimization (GRPO) (Shao et al., 2024) is widely adopted as it eliminates this need for a value model (Schulman et al., 2017) by replacing it with a simpler Monte Carlo estimation: Given a prompt $x$, GRPO samples a group of $G$ candidate completions $\{y^{(i)}\}_{i=1}^G$ from the old policy $\pi_{\theta_{\text{old}}}$. Instead of estimating the baseline with a learned value function, it computes the relative advantage of each sample as its reward minus the group mean reward (Liu et al., 2025d). Incorporating a KL penalty, the resulting optimization objective is:

$$\mathcal{J}_{\text{GRPO}}(\pi_\theta) = \mathbb{E}_{x \sim \mathcal{D}, y^{(1)}, \dots, y^{(G)} \sim \pi_{\theta_{\text{old}}}(\cdot|x)}$$

$$\left[ \frac{1}{G} \sum_{i=1}^G \frac{1}{|y^{(i)}|} \sum_{k=1}^{|y^{(i)}|} \min(\rho^{k,(i)}\hat{A}^{(i)}, \text{clip}(\rho^{k,(i)}, 1-\epsilon, 1+\epsilon)\hat{A}^{(i)}) - \beta D_{\text{KL}}(\pi_\theta, \pi_{\text{ref}}) \right], \quad (4)$$

where $\rho^{k,(i)} = \frac{\pi_\theta(y^{k,(i)}|x, y^{<k,(i)})}{\pi_{\theta_{\text{old}}}(y^{k,(i)}|x, y^{<k,(i)})}$ is the token-level importance ratio between policies, and $\hat{A}^{(i)} = R(x, y^{(i)}) - \frac{1}{G}\sum_{j=1}^G R(x, y^{(j)})$ denotes the group-relative advantage. A more comprehensive introduction to reinforcement learning is provided in Appendix B.1.

## 3 THE CHALLENGE OF THE TOKEN-LEVEL PERSPECTIVE IN dLLMs

The core challenge in applying reinforcement learning to dLLMs stems from a fundamental mismatch between the dLLMs' probabilistic modeling and the assumptions inherent in standard RL algorithms. Mainstream policy gradient algorithms, including GRPO as formulated in Eq. (4), are intrinsically designed for autoregressive models that factorize the sequence likelihood into a product of conditional probabilities as $\pi_\theta(y|x) = \prod_{k=1}^L \pi_\theta(y^k|x, y^{<k})$. This factorization naturally defines a sequence of actions, allowing the objective to assign rewards at the token level via the importance ratio $\frac{\pi_\theta(y^k|x, y^{<k})}{\pi_{\theta_{\text{old}}}(y^k|x, y^{<k})}$. However, dLLMs generate text non-autoregressively, refining a complete sequence over iterative denoising steps. Consequently, the autoregressive conditional probability $\pi_\theta(y^k|x, y^{<k})$ is ill-defined or hard to compute (see Appendix B.2 for detailed discussions), forcing existing methods to rely on heuristic proxies to bridge this gap.

Early attempts to resolve this incompatibility focused on finding a suitable token-level substitute for the AR conditional probability $\pi_\theta(y^k|x, y^{<k})$. For instance, d1 (Zhao et al., 2025) employed a mean-field approximation $\log p_\theta(y^k|x)$ as a proxy for $\log \pi_\theta(y^k|x, y^{<k})$. This approach is inaccurate as it ignores the context provided by other tokens in the sequence $y$. Recognizing this limitation, subsequent work such as UniGRPO (Yang et al., 2025) and Coupled-GRPO (Gong et al., 2025) proposed a more sophisticated proxy: the token's contribution to the ELBO, $\mathcal{L}_\theta^k(y|x)$[1]:

$$\mathcal{L}_\theta^k(y|x) \triangleq \mathbb{E}_{t \sim \mathcal{U}[0,1]} \mathbb{E}_{y_t \sim q_t(y_t|y,x)} \left[ \frac{1}{t} \mathbf{1}[y_t^k = \text{M}] \log p_\theta(y^k|y_t, x) \right]. \quad (5)$$

The $\mathcal{L}_\theta^k(y|x)$ is better aligned with the nature of dLLM generation, since its computation involves predicting a masked token given both $x$ and the surrounding unmasked context from $y$. However, the ELBO, $\mathcal{L}_\theta(y|x) = \sum_{k=1}^L \mathcal{L}_\theta^k(y|x)$, is only valid on sequence level as lowerbound of $\log \pi_\theta(y|x)$. An individual component $\mathcal{L}_\theta^k(y|x)$ has no formal probabilistic interpretation as a conditional likelihood. Therefore, the decomposition of ELBO at the token level and its heuristic substitution into the GRPO objective (Eq. (4)) introduces an unknown inconsistency.

---

[1]Implementations may vary; for instance, UniGRPO's ELBO-like term omits the $\frac{1}{t}$ coefficient from Eq. (5), akin to the simplified objective in DDPM (Ho et al., 2020).

As analyzed above, the core issue is not merely about finding a better token-level proxy, but that the token-level decomposition itself fundamentally does not fit for dLLMs. Forcing a dLLM into a token-level AR framework rests on an improper assumption. This motivates our approach: instead of adapting the dLLM model to fit the algorithm, we must adapt the algorithm to respect the holistic, sequence-level nature of the dLLM model.

# 4 A PRINCIPLED SEQUENCE-LEVEL RL FRAMEWORK FOR dLLMS

Motivated by analysis in Section 3, we propose *ELBO-based Sequence-level Policy Optimization (ESPO)* algorithm, which is tailored for dLLMs. Our approach is built on a sequence-level action space, uses the ELBO as a tractable proxy for the sequence log-likelihood, and incorporates crucial stabilization techniques for both the policy gradient objective and the KL-divergence regularizer.

## 4.1 THE SEQUENCE-LEVEL POLICY OBJECTIVE WITH ELBO APPROXIMATION

We begin by reformulating the RL objective to align with the nature of dLLM generation.

**Sequence-Level Objective.** Instead of viewing each token as an independent action, we treat the generation of the entire sequence $y$ as a single, atomic action. This naturally leads to a sequence-level adaptation of the GRPO framework (Eq. (4)), where the token-level summation is removed. The objective now depends on a sequence-level importance ratio $\frac{\pi_\theta(y^{(i)}|x)}{\pi_{\theta_{old}}(y^{(i)}|x)}$. By substituting the intractable log-likelihood $\log \pi(y^{(i)}|x)$ with its ELBO approximation $\mathcal{L}(y^{(i)}|x)$, we obtain the sequence-level ratio for dLLM:

$$\rho_{\text{seq}}^{(i)} = \frac{\exp \mathcal{L}_\theta(y^{(i)}|x)}{\exp \mathcal{L}_{\theta_{old}}(y^{(i)}|x)} = \exp(\mathcal{L}_\theta(y^{(i)}|x) - \mathcal{L}_{\theta_{old}}(y^{(i)}|x)) \tag{6}$$

Plugging this into the GRPO objective gives us a vanilla sequence-level objective[2] :

$$\mathcal{J}_{\text{seq}}(\pi_\theta) = \mathbb{E}_{x\sim\mathcal{D}, y^{(1)},...,y^{(G)}\sim\pi_{\theta_{old}}(\cdot|x)} \left[ \frac{1}{G} \sum_{i=1}^{G} \min(\rho_{\text{seq}}^{(i)}\hat{A}^{(i)}, \text{clip}(\rho_{\text{seq}}^{(i)}, 1-\epsilon, 1+\epsilon)\hat{A}^{(i)}) \right], \tag{7}$$

While this sequence-level formulation correctly avoids the pitfall of splitting the ELBO at the token level, we found this vanilla formulation to be practically unusable. The magnitude of the raw ELBO difference, $\mathcal{L}_\theta(y^{(i)}|x) - \mathcal{L}_{\theta_{old}}(y^{(i)}|x)$, typically scales linearly with the sequence length $L$. The subsequent exponentiation results in astronomically large or infinitesimally small ratios, causing unstable optimization.

To address this instability, we draw inspiration from GSPO (Zheng et al., 2025) and normalize the log-ratio by the sequence length $L$. This transforms the unstable, raw log-likelihood difference into a stable, per-token scale. Our final, stabilized importance ratio is:

$$\rho_{\text{seq}}^{(i)} = \exp\left(\frac{1}{L}(\mathcal{L}_\theta(y^{(i)}|x) - \mathcal{L}_{\theta_{old}}(y^{(i)}|x))\right) = \exp\left(\frac{1}{L}\sum_{k=1}^{L}(\mathcal{L}_\theta^k(y^{(i)}|x) - \mathcal{L}_{\theta_{old}}^k(y^{(i)}|x))\right). \tag{8}$$

Plugging this stabilized ratio into the sequence-level objective (Eq. (7)) enables effective training.

**Empirical Validation.** To validate our design choices of using a sequence-level action space with an ELBO approximation, we conduct a complete ablation study on the Sudoku benchmark. We compare four variants: token-level actions with a mean-field likelihood, token-level with ELBO, sequence-level with mean-field, and our proposed sequence-level with ELBO. The precise mathematical formulations are detailed in Appendix C.1. As shown in Fig. 1, the results provide strong empirical support for our analysis:

---

[2]The KL term is omitted here for simplicity; a detailed discussion is provided later in Section 4.2.

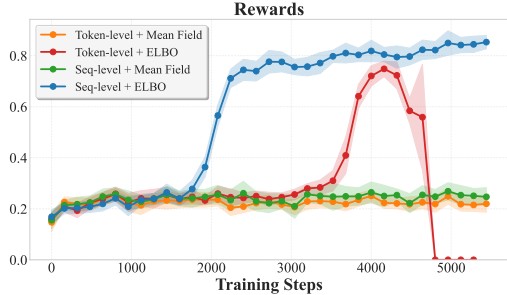 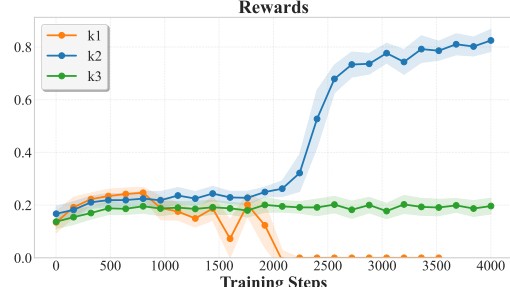

Figure 1: **Training performance on the Sudoku task under different action space (*Token-level vs. Sequence-level*) and likelihood approximations (*Mean-field vs. ELBO*).** Our method (blue) combines a sequence-level action space with an ELBO approximation, yielding the most stable and highest performance.

Figure 2: **Training performance on the Sudoku task with different KL-divergence estimators.** The $k_2$ estimator (blue) achieves stable and superior performance. The $k_1$ estimator (orange) is highly unstable and collapses, while the $k_3$ estimator (green) stagnates.

- ***Mean-field is a poor proxy.*** Both mean-field variants (orange and green curve) fail to learn effectively, confirming that this approximation is fundamentally misaligned with the conditional denoising process.

- ***Token-level ELBO is unstable.*** The Token-level + ELBO approach (red curve), while initially promising, suffers from high instability and eventual collapse. This highlights the inconsistency of breaking the ELBO's integrity across tokens.

- ***Sequence-level ELBO is superior.*** Our proposed method (blue curve) is the only one to achieve fast, stable learning that converges to the highest reward, validating that the sequence-level action space paired with the holistic ELBO proxy is the correct and most effective approach.

## 4.2 STABLE KL-DIVERGENCE ESTIMATION

The complete GRPO objective includes a KL-divergence term to regularize policy updates against a reference policy. A common choice for token-level autoregressive models is the $k_3$ estimator (Schulman, 2017). However, a direct application of the $k_3$ estimator to our sequence-level objective is highly problematic. Its formulation, when approximating log-likelihoods with the ELBO, is:

$$\widehat{\mathbb{KL}}_{k3} = \exp\Big(\mathcal{L}_{\text{ref}}(y^{(i)}|x) - \mathcal{L}_\theta(y^{(i)}|x)\Big) - 1 - \Big(\mathcal{L}_{\text{ref}}(y^{(i)}|x) - \mathcal{L}_\theta(y^{(i)}|x)\Big). \quad (9)$$

As the formula shows, the $k_3$ estimator contains an exponential term, which reintroduces the unstable problem similar to Eq. (6) at the sequence level. To circumvent this exponential instability, we adopt the more robust $k_2$ estimator (Schulman, 2017), which is known to yield a correct gradient for KL optimization (Tang & Munos, 2025b). Our practical and stable KL estimate becomes:

$$\widehat{\mathbb{KL}}_{k2} = \tfrac{1}{2}\Big(\mathcal{L}_\theta(y^{(i)}|x) - \mathcal{L}_{\text{ref}}(y^{(i)}|x)\Big)^2. \quad (10)$$

Unlike the $k_3$ estimator, the $k_2$ estimator is a simple quadratic function of the ELBO difference. This polynomial form avoids the exponential term entirely, ensuring that the gradient signal from the KL regularizer remains stable and well-behaved, even for long sequences.

**Empirical Validation.** To demonstrate the critical impact of the KL estimator, we conduct an ablation study on the Sudoku task comparing the performance of the $k_1$, $k_2$, and $k_3$ estimators. More details can be referenced in Appendices C.2 and G.3. As shown in Fig. 2, the choice of estimator is crucial for stable learning. The $k_3$ estimator (green) fails to learn, with rewards stagnating at a low level, which is consistent with our analysis of its unstable property. The $k_1$ estimator (orange) is also highly unstable; while it shows some initial progress, its performance violently fluctuates before collapsing to zero midway through training. In stark contrast, the $k_2$ estimator (blue) enables

Table 1: **Model performance on mathematics and planning benchmarks.** For each task, we train a separate model. Countdown results with [†] for LLaDA, diffu-GRPO, and wd1 are from Zhao et al. (2025); Tang et al. (2025), while other results are reproduced as detailed in Section 5.1. $\Delta$ denotes the improvement of ESPO over LLaDA or Dream model without reinforcement post-training.

| Model / Seq Len | GSM8K(0) | | | | MATH(0) | | | | Countdown | | | | Sudoku | | | |
| --- | --- | --- | --- | --- | --- | --- | --- | --- | --- | --- | --- | --- | --- | --- | --- | --- |
| | 128 | 256 | 512 | Avg. | 128 | 256 | 512 | Avg. | 128 | 256 | 512 | Avg. | 128 | 256 | 512 | Avg. |
| LLaDA | 71.3 | 76.2 | 80.2 | 75.9 | 34.4 | 35.2 | 41.4 | 37.0 | 20.7[†] | 19.5[†] | 16.0[†] | 18.7[†] | 24.8 | 16.2 | 6.0 | 15.7 |
| + diffu-GRPO(d1) | 74.6 | 78.1 | 81.2 | 78.0 | 34.9 | 36.6 | 41.7 | 37.7 | 33.2[†] | 31.3[†] | 37.1[†] | 33.9[†] | 26.7 | 24.1 | 15.9 | 22.2 |
| + wd1 | 77.2 | 80.8 | 82.3 | 80.1 | 33.3 | 37.7 | 39.8 | 36.9 | 47.7 | 51.2[†] | 46.1[†] | 48.3 | 22.6 | 22.0 | 24.6 | 23.1 |
| + ESPO (ours) | 80.0 | 82.3 | 83.7 | 82.0 | 36.0 | 39.0 | 43.4 | 39.5 | 81.6 | 82.0 | 79.3 | 81.0 | 92.7 | 84.7 | 80.5 | 86.0 |
| $\Delta$ | +8.7 | +6.1 | +3.5 | +6.1 | +1.6 | +3.8 | +2.0 | +2.5 | +60.9 | +62.5 | +63.3 | +62.3 | +67.9 | +68.5 | +74.5 | +70.3 |
| Dream | 75.8 | 81.3 | 80.7 | 79.3 | 38.2 | 45.7 | 48.0 | 44.0 | 8.5 | 7.8 | 17.4 | 11.2 | 9.3 | 2.1 | 14.0 | 8.5 |
| + diffu-GRPO(d1) | 77.0 | 81.9 | 81.7 | 80.2 | 39.4 | 46.9 | 48.9 | 45.1 | 27.3 | 27.7 | 37.5 | 30.8 | 64.4 | 69.7 | 51.1 | 61.7 |
| + wd1 | 76.3 | 82.4 | 82.9 | 80.5 | 39.5 | 47.4 | 50.5 | 45.8 | 28.9 | 37.9 | 42.2 | 36.3 | 29.5 | 39.2 | 30.3 | 33.0 |
| + ESPO (ours) | 79.6 | 82.3 | 82.0 | 81.3 | 40.3 | 47.4 | 50.3 | 46.0 | 68.8 | 66.8 | 64.8 | 66.8 | 71.7 | 72.3 | 71.3 | 71.8 |
| $\Delta$ | +3.8 | +1.0 | +1.3 | +2.0 | +2.1 | +1.7 | +2.3 | +2.0 | +60.3 | +59.0 | +47.4 | +55.6 | +62.4 | +70.2 | +57.3 | +63.3 |

stable and efficient learning, consistently improving and ultimately converging to the highest reward. This result empirically validates that the $k_2$ estimator is the most robust and effective choice for our sequence-level framework.

# 5 EXPERIMENT

## 5.1 EXPERIMENTAL SETUP

**Models & Tasks** We apply our ESPO algorithm to two open-source dLLMs: LLaDA-8B-Instruct (Nie et al., 2025) and Dream-7B-Instruct (Ye et al., 2025). For reference, we also report evaluation results on LLaDA-1.5 for comparison. Following prior work (Zhao et al., 2025; Tang et al., 2025), we focus on three categories of reasoning tasks: (i) Mathematics: GSM8K (Cobbe et al., 2021a) and MATH (Hendrycks et al., 2021), (ii) Coding: HumanEval (Chen et al., 2021) and MBPP (Austin et al., 2021b), EvalPlus (HumanEval+ and MBPP+) (Liu et al., 2023a) and (iii) Planning: Countdown and Sudoku (Ye et al., 2024). For mathematical tasks, we train on the official training split of each dataset. For coding tasks, we follow Gong et al. (2025) and train on a subset of AceCoder-87K (Zeng et al., 2025). For planning tasks (Countdown and Sudoku), we train on synthetic training data following Zhao et al. (2025).

**Training** To evaluate ESPO's effectiveness, we apply it directly to the models without additional task-specific SFT. For stable training, we adopt two standard variance reduction techniques: following Zhu et al. (2025), we use antithetic sampling, which shares the same noise level and mask positions when estimating ELBO differences; and, inspired by Gong et al. (2025), we employ a coupled-sampling scheme. All experiments use 2 Monte Carlo samples and a policy update value of $\mu = 8$. Additional details on variance-reduction strategies are provided in Appendix D, while further training configurations and extended results can be found in Appendices E and G.

**Evaluation** Following d1, we evaluate all benchmarks at generation lengths of 128, 256, and 512. For mathematics and coding tasks, we use the official evaluation scripts provided by the LLaDA and Dream repositories, respectively. For planning tasks, we use the evaluation code based on d1. To ensure fair comparison, we re-run those baselines whose reported settings differ from ours or whose results are missing for specific lengths (e.g., when length-128 results were not originally reported).

## 5.2 BENCHMARK RESULTS

As shown in Table 1 and Table 2, our sequence-level RL algorithm, ESPO, consistently and significantly outperforms both the original models and prior token-level RL baselines like diffu-GRPO (Zhao et al., 2025) and wd1 (Tang et al., 2025), while achieving performance on coding tasks that is comparable to LLaDA-1.5. Notably, although our models are trained only on the sequence length of 256, the improvements generalize effectively to other lengths.

Table 2: **Model performance on coding benchmarks.** We train a single model and evaluate it across multiple coding benchmarks (HumanEval and MBPP) at different sequence lengths. $\Delta$ denotes the improvement of ESPO over LLaDA model without reinforcement post-training. ESPO consistently enhances the performance while even achieving competitive results compared with LLaDA-1.5, which was trained on a privately collected dataset at a significantly larger scale.

| | HumanEval(0) | | | | | | | | MBPP(3) | | | | | | | |
| | - | | | | Plus | | | | - | | | | Plus | | | |
| Model / Seq Len | 128 | 256 | 512 | Avg. | 128 | 256 | 512 | Avg. | 128 | 256 | 512 | Avg. | 128 | 256 | 512 | Avg. |
|---|---|---|---|---|---|---|---|---|---|---|---|---|---|---|---|---|
| **LLaDA** | 26.8 | 37.8 | 48.8 | 37.8 | 23.2 | 30.5 | 41.5 | 31.7 | 38.2 | 37.0 | 38.2 | 37.8 | 36.8 | 36.8 | 37.8 | 37.1 |
| + diffu-GRPO(d1) | 25.6 | 36.0 | **50.0** | 37.2 | 22.0 | 29.9 | 37.2 | 29.7 | 34.8 | 36.6 | 38.0 | 36.5 | 31.8 | 34.8 | 39.4 | 35.3 |
| + wd1 | **34.7** | 38.4 | 38.4 | 37.2 | **29.9** | 29.9 | 32.9 | 30.9 | 38.0 | 37.2 | 34.4 | 36.5 | 31.8 | 32.8 | 35.6 | 33.4 |
| + ESPO (ours) | 28.1 | **42.1** | **50.0** | **40.1** | 24.4 | **36.6** | 42.7 | **34.6** | **47.4** | **44.6** | **44.2** | **45.4** | **38.9** | **41.6** | **42.6** | **41.0** |
| $\Delta$ | +1.3 | +4.3 | +1.2 | +2.3 | +1.2 | +6.1 | +1.2 | +2.9 | +9.2 | +7.6 | +6.0 | +7.6 | +2.1 | +4.8 | +4.8 | +3.9 |
| LLaDA-1.5 | 29.3 | 39.6 | 51.9 | 40.3 | 23.2 | 32.3 | 45.1 | 33.5 | 39.6 | 39.9 | 38.8 | 39.4 | 38.8 | 40.4 | 37.3 | 38.8 |

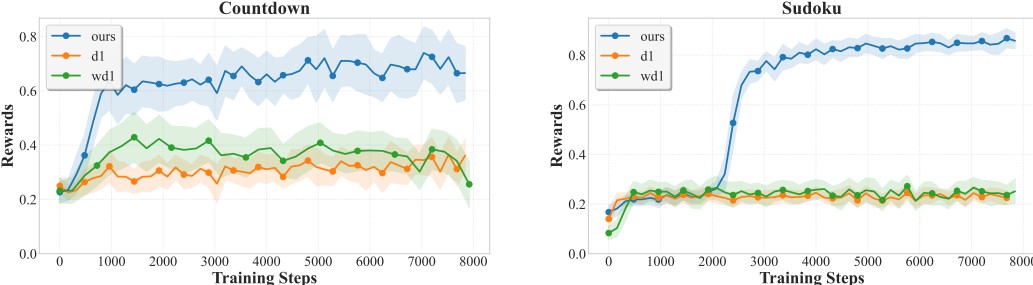

Figure 3: **Training dynamics with different methods for Countdown and Sudoku tasks on LLaDA-8b-Instruct.**

**Dominant Performance on Planning Tasks.** The most dramatic improvements remain in planning tasks. On Countdown, ESPO consistently outperforms the strongest baselines by 20–40 absolute points, while on the Sudoku tasks, the gains are up to over 60 points depending on the sequence length. This strongly validates our core hypothesis: a sequence-level objective is superior for tasks requiring holistic consistency, a property that token-level optimization fails to capture effectively.

**Consistent Gains on Mathematics and Coding.** On established mathematics and coding benchmarks, the gains are more modest but still consistently positive. Note that the pre-existing knowledge of the base models acts as a performance ceiling, limiting the maximum achievable gains through RL fine-tuning alone. Despite this, our method reliably enhances performance, surpassing all previous token-level dLLM-RL methods averaged over sequence lengths, and comparable with LLaDA-1.5. This demonstrates the broad effectiveness of our approach even on knowledge-intensive tasks.

## 5.3 TRAINING DYNAMICS

Fig. 3 shows the training reward dynamics on Countdown and Sudoku, highlighting our method's superior performance. The difference is most pronounced on Sudoku, where our sequence-level method rapidly converges to a near-optimal policy, while the token-level d1 and wd1 baselines completely stagnate at low rewards. Similar trends can be observed on the Countdown task.

## 5.4 ABLATION EXPERIMENTS

**Number of Monte Carlo Samples** We investigate the impact of the number of Monte Carlo (MC) samples used to estimate the ELBO. For this ablation, all other hyperparameters match those in our main experiments. As shown in Fig. 4, increasing the number of MC samples consistently improves training stability, but the magnitude of the effect varies by task. For signal-rich tasks such as Sudoku, using more MC samples substantially accelerates convergence—MC=1 converges slowly, while MC=2 or MC=4 leads to noticeably faster progress. In contrast, for sparse-reward tasks such as Countdown, the benefits of additional MC samples are much smaller.

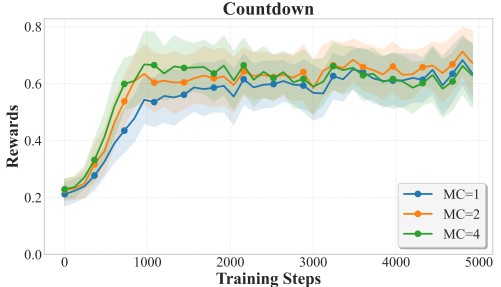 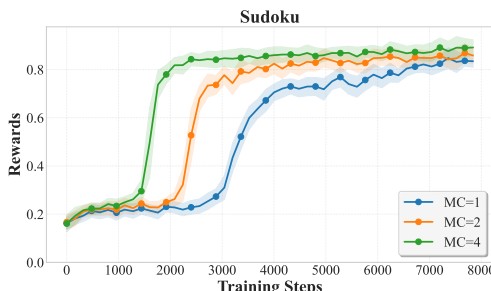

Figure 4: **Ablation study on the number of Monte Carlo samples for Countdown and Sudoku.** We evaluate training performance with different MC sample counts (1, 2, 4), showing the effect of increased sampling on reward optimization.

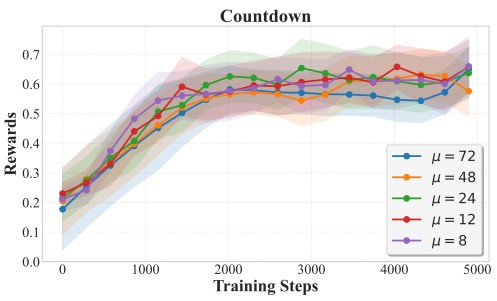 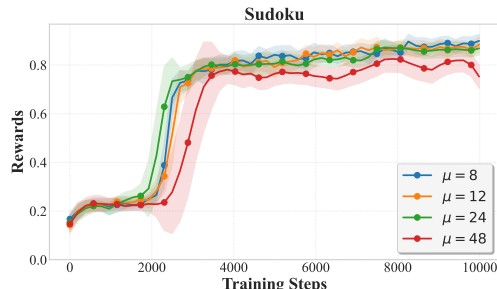

Figure 5: **Ablation study on the policy update values ($\mu$) for Countdown and Sudoku.** The reward curves illustrate performance across a range of $\mu$ values. While smaller values (e.g., 8, 12) lead to faster initial convergence on Sudoku, the method is robust and achieves similarly high rewards across all settings for Countdown and Sudoku tasks.

**Number of different policy update values $\mu$** In these experiments, we varied $\mu$ while keeping all other settings identical to our main setup. Our method demonstrates remarkable robustness to the number of policy updates ($\mu$) per data collection phase, as shown in Fig. 5. On both Countdown and Sudoku, our approach successfully scales to large $\mu$ values, converging to a similarly high reward across all settings, though smaller values (e.g., 8, 12) exhibit faster initial convergence on Sudoku. We hypothesize that this robustness is linked to task complexity. While the policy for relatively simple planning tasks can be updated aggressively, more complex domains like mathematics and coding may have more brittle reward landscapes. In such cases, a large $\mu$ could lead to instability, making smaller, more conservative update frequencies a safer and more effective choice.

## 5.5 DISCUSSION OF TRAINING EFFICIENCY

In reinforcement learning for dLLMs, the total training time can be roughly decomposed into two components: data generation (sampling) and policy updates (training). For dLLMs, the generation phase typically dominates: each step requires non-autoregressive denoising over the full sequence, preventing extensive reuse of KV cache. In contrast, the number of Monte Carlo samples $M$ used for ELBO estimation only affects the policy update phase, which is less computationally intensive.

To quantify the computational cost, we use FLOPs as a hardware-agnostic proxy. For a model with $N$ parameters and sequence length $D$, the forward and backward passes approximately require $C_{\text{forward}} = 2ND$ and $C_{\text{backward}} = 4ND$ FLOPs (Kaplan et al., 2020), respectively. We obtain the total FLOPS per sample with coupled sampling as $2ND(K + 6\mu M)$ (see Appendix F for details), where $K$ is the sampling step, $\mu$ is the policy update values, and $M$ is the number of MC samples.

We further validate this analysis on coding tasks using the same training configuration as in Section 5.1, except that the number of MC samples $M$ is varied from 1, 2, and 4. The resulting theoretical FLOPs and empirical wall-clock training time are summarized in Table 3.

Table 3: **Training cost under different numbers of Monte Carlo (MC) samples for 100 steps** on the coding task. Following the setting in Section 5.1, the training parameters are set as denoising steps $K = 256$ and policy updates value $\mu = 8$. Experiments are conducted under H200 GPUs.

| MC Samples ($M$) | 1 | 2 | 4 |
|---|---|---|---|
| Wall-clock Time (hrs) | 5.61 (100%) | 6.78 (121%) | 9.06 (161%) |
| Theoretical FLOPs | $608ND$ (100%) | $704ND$ (116%) | $896ND$ (147%) |

First, we observe that the theoretical FLOPs capture the overall trend of wall-clock time, consistent with the fact that dLLM inference is a compute-bound process. The small discrepancy may be attributed to factors such as GPU utilization, memory bandwidth, and communication overhead.

Second, focusing on the relative growth as $M$ increases, our approach shows a much more moderate increase in training cost. Since generation costs are fixed (dominated by $K$), increasing $M$ only adds to the policy update term, which is minor when $M$ is within a proper range. For example, increasing $M$ from 1 to 4 raises the total FLOPs by only about 47% in our case, consistent with the observed wall-clock time. In contrast, for ELBO-based DPO algorithms such as VRPO (Zhu et al., 2025), their training time scales almost linearly with $M$ (e.g., a $4\times$ increase from $M = 1$ to $M = 4$).

## 6 RELATED WORK

**Diffusion language models.** dLLMs have recently emerged as a powerful approach for sequence modeling, leveraging discrete diffusion (Austin et al., 2021a; Campbell et al., 2022; Meng et al., 2023; Lou et al., 2024; Zhao et al., 2024), or in particular masked diffusion models (Ou et al., 2024; Sahoo et al., 2024; Shi et al., 2024). Distinct from autoregressive models (OpenAI, 2023) that heavily rely on the left-to-right causal factorization of the sequence, dLLMs instead operate directly on the sequence-level for each diffusion step and enable flexible order sampling (Kim et al., 2025) and parallel decoding (Arriola et al., 2025). While they have shown great promise in various domains (Kwon et al., 2025; Ma et al., 2025; Liu et al., 2025c; Wu et al., 2025; Hu et al., 2025; You et al., 2025b) when scaled up (Nie et al., 2024; Prabhudesai et al., 2025; Ni & the team, 2025), and their adaptation to downstream tasks with specific reward signals still has room for exploration.

**RL for language models.** Reinforcement learning (Schulman et al., 2017) has proven effective in enhancing language model performance, particularly when rewards can be obtained via automated verifiers, a paradigm known as reinforcement learning with verifiable rewards (RLVR) (OpenAI, 2024). Methods such as GRPO (Shao et al., 2024; Guo et al., 2025) and related works (Luo et al., 2025) have improved reasoning capabilities and achieved strong performance across diverse tasks. However, directly applying these techniques to dLLMs is challenging, primarily due to the difficulty of likelihood computation.

**RL for diffusion language models.** Except for continuous diffusion models (Wallace et al., 2024; Black et al., 2024; Liu et al., 2025a; McAllister et al., 2025), several works have explored leveraging RL to enhance the performance of diffusion language models. For general discrete diffusion models, Zhang et al. (2025) proposes a framework by target concrete score matching, Venkatraman et al. (2024) proposes Relative Trajectory Balance to achieve unbiased sampling of posterior distribution, and Zekri & Boullé (2025) proposes score entropy policy optimization. For masked diffusion models, LLadou (Huang et al., 2025) uses RL and trains an additional module to predict the position for decoding. It models each denoising step as an action, stores the entire trajectory, and optimizes the conditional probability of each intermediate state. However, in offline settings, optimizing over multi-step trajectories requires repeated forward and backward passes growing linearly with the number of denoising steps, which is prohibitive for large-scale models. Various works (Zhao et al., 2025; Tang et al., 2025) have proposed mean-field surrogates of the likelihood that are efficient to compute to enhance efficiency. Yet, the approximations significantly trade off with quality and hinder the multi-step generation capability of the model. Our work instead develops a discrete diffusion RL objective from a principled sequence-level likelihood evaluation via efficient MC estimate, different from Yang et al. (2025); Gong et al. (2025) that evaluates the likelihood on the token-level heuristics.

# 7 CONCLUSION

In this work, we identified the fundamental incompatibility between autoregressive RL objectives and the non-autoregressive nature of dLLMs, and proposed ESPO, a principled sequence-level reinforcement learning framework that leverages the ELBO as a tractable proxy for sequence likelihood. By treating sequence generation as a single action and introducing stabilized importance ratios and KL regularization, ESPO eliminates inconsistencies in prior token-level approaches and enables robust and efficient large-scale training. Extensive experiments on math, coding, and planning benchmarks demonstrate that our method consistently outperforms existing dLLM-RL baselines. Our results establish sequence-level optimization as a principled and empirically effective paradigm for RL in diffusion language models.

## ACKNOWLEDGMENTS

This work was supported by the Beijing Major Science and Technology Project under Contract no. Z251100008425002; the National Natural Science Foundation of China (Nos. 62522609, 92470118); the Beijing Natural Science Foundation (No. L247030); and the fund for building world-class universities (disciplines) of Renmin University of China.

Personally, we thank Jiajun Liu and Shusheng Xu for their generous assistance throughout this project.

## ETHICS STATEMENT

This work is purely methodological and does not involve human subjects or sensitive data. Experiments are conducted on publicly available benchmarks, and results are reported in aggregate. We highlight that while our techniques improve efficiency and accuracy, they should be applied responsibly to avoid potential misuse.

## REPRODUCIBILITY STATEMENT

We have open-sourced our code in `https://github.com/ML-GSAI/ESPO`. Comprehensive explanation and details of our theory and experiments can be found in Appendices C to E.

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

## A  THE USE OF LARGE LANGUAGE MODELS

During the writing of this work, Large Language Models (LLMs) were used as auxiliary tools to assist with language polishing, grammar checking, and improving the readability of tables and figure captions. We will take full responsibility for all parts of the paper.

## B  FROM GENERIC RL PRINCIPLES TO DLLM RL FORMULATIONS

**Scope of Derivation:** We focus on the output reward setting, where the reward $R(x, y)$ is evaluated only upon the completion of the entire sequence $y$. This is the mainstream setup for reasoning tasks (e.g., Math, Coding) in post-training.

### B.1  GENERIC RL OBJECTIVE AND GRADIENT DERIVATIONS

The standard RL objective is defined to maximize the expected accumulated reward :

$$\mathcal{J}(\pi_\theta) = \mathbb{E}_{y \sim \pi_\theta(\cdot|x)} \left[ R(x, y) \right]. \tag{11}$$

Using the log-derivative trick and introducing Importance Sampling (IS) to evaluate updates under the behavior policy $\pi_{\text{old}}$, the gradient can be derived as below (Williams, 1992):

$$\nabla_\theta \mathcal{J}(\pi_\theta) = \nabla_\theta \mathbb{E}_{y \sim \pi_{\text{old}}(\cdot|x)} \left[ \frac{\pi_\theta(y|x)}{\pi_{\text{old}}(y|x)} R(x, y) \right], \tag{12}$$

$$= \mathbb{E}_{y \sim \pi_{\text{old}}(\cdot|x)} \left[ \frac{\pi_\theta(y|x)}{\pi_{\text{old}}(y|x)} \nabla_\theta \log \pi_\theta(y|x) R(x, y) \right], \tag{13}$$

$$= \mathbb{E}_{y \sim \pi_{\text{old}}(\cdot|x)} \left[ \underbrace{\frac{\pi_\theta(y|x)}{\pi_{\text{old}}(y|x)}}_{\text{IS Ratio}} \nabla_\theta \log \pi_\theta(y|x) A(x, y) \right], \tag{14}$$

where $A(x, y)$ denotes the advantage function (typically $R(x, y) - b(x)$) introduced for variance reduction.

For Autoregressive models, the sequence likelihood factorizes as $\pi(y|x) = \prod_{i=1}^{d} \pi(y^i|x, y^{<i})$. Substituting this into Eq. (14) yields the full sequence-level gradient:

$$\mathbb{E}_{y \sim \pi_{\text{old}}} \left[ \underbrace{\frac{\prod_{i=1}^{d} \pi_\theta(y^i|x, y^{<i})}{\prod_{i=1}^{d} \pi_{\text{old}}(y^i|x, y^{<i})}}_{\text{unstable}} \sum_{i=1}^{d} \nabla_\theta \log \pi_\theta(y^i|x, y^{<i}) A(x, y) \right]. \tag{15}$$

The cumulative product term (in blue) introduces severe variance and numerical instability (exploding or vanishing gradients) as $d$ increases, rendering Eq. (15) impractical for long sequences.

To address this, Token-level RL methods (e.g., PPO (Schulman et al., 2017), GRPO (Shao et al., 2024)) approximate the full ratio by decomposing it into per-token ratios. While practical implementations utilize clipping mechanisms or KL penalties to constrain policy updates and ensure the validity of this approximation, we omit these regularizers here to focus on the core gradient structure:

$$\nabla \mathcal{J}_{\text{PPO/GRPO}} = \mathbb{E}_{y \sim \pi_{\text{old}}} \left[ \frac{1}{d} \sum_{i=1}^{d} \frac{\pi_\theta(y^i|x, y^{<i})}{\pi_{\text{old}}(y^i|x, y^{<i})} \nabla_\theta \log \pi_\theta(y^i|x, y^{<i}) A(x, y) \right]. \tag{16}$$

Conversely, Sequence-level RL methods (e.g., GSPO (Zheng et al., 2025)) argue that applying off-policy correction at the token level for a sequence-level reward is heuristic. They retain the sequence-level perspective but stabilize the ratio via length normalization. Similarly, omitting clipping and KL terms for simplicity, the gradient is formulated as:

$$\nabla \mathcal{J}_{\text{GSPO}} = \mathbb{E}_{y \sim \pi_{\text{old}}} \left[ \frac{1}{d} \left( \frac{\prod_{i=1}^{d} \pi_\theta(y^i|x, y^{<i})}{\prod_{i=1}^{d} \pi_{\text{old}}(y^i|x, y^{<i})} \right)^{\frac{1}{d}} \sum_{i=1}^{d} \nabla_\theta \log \pi_\theta(y^i|x, y^{<i}) A(x, y) \right]. \quad (17)$$

## B.2 Adapting RL Objectives to Diffusion Language Models

Applying the derived principles to dLLMs involves a critical choice regarding the action space and likelihood approximation.

The most direct approach is to formulate the RL problem at the *trajectory level* as a multi-step Markov decision process, where each denoising step $t$ constitutes an action (Huang et al., 2025). Here, a "trajectory" $(y_T, y_{T-1}, \ldots, y_0)$ represents the evolution from pure noise $y_T$ to clean data $y_0$. In this case, we factorizes over diffusion timesteps $t$ as $\prod_{t=T}^{1} \pi_\theta(y_{t-1}|x, y_t)$ rather than token index $i$ as $\prod_{i=1}^{d} \pi(y^i|x, y^{<i})$.

In this view, the RL gradient involves summing over all $T$ timesteps, where $T$ equals the number of sampling steps (typically $T \in [50, 1000]$). Taking the simplest online policy gradient as an example:

$$\nabla \mathcal{J}_{\text{Traj}} = \mathbb{E}_{\pi_\theta} [\underbrace{\sum_{t=1}^{T} \nabla_\theta \log \pi_\theta(y_{t-1}|y_t, x)}_{\text{Requires } T \text{ gradient evaluations}} \cdot R(x, y_0)]. \quad (18)$$

While theoretically sound, this approach suffers from a prohibitive computational bottleneck due to the nature of the generative process. **Crucially, unlike Autoregressive models where all conditional probabilities $\log \pi(y^i|y^{<i})$ can be computed in parallel via a single network forward pass (enabled by causal masking), dLLMs fundamentally require $T$ sequential network evaluations to compute the full trajectory probability.** Consequently, calculating gradients for $\nabla \mathcal{J}_{\text{Traj}}$ requires backpropagating through $T$ separate forward passes for every sample. This scales the compute cost **linearly with $T$** (e.g., $50 \times \sim 1000 \times$ more expensive than a single step), rendering it infeasible for large-scale post-training.

Given this constraint, we can optimize the final generated sequence directly using the ELBO as a proxy for the intractable log-likelihood. This is computationally feasible because the ELBO can be efficiently estimated via Monte Carlo (MC) sampling, requiring only $K$ forward passes. In practice, the MC sample size is typically small ($K \approx 2 \sim 4$), offering a direct trade-off between the quality of the likelihood proxy and computational efficiency. This leads to the fundamental divergence between token-level and sequence-level formulations:

- **Sequence-Level Validity:** The ELBO $\mathcal{L}_\theta(y|x)$ is a mathematically rigorous variational lower bound for the joint sequence log-likelihood: $\mathcal{L}_\theta(y|x) \leq \log \pi_\theta(y|x)$. Therefore, substituting $\mathcal{L}_\theta(y|x)$ into the Sequence-level gradient formulation (Eq. (17)) preserves the mathematical integrity of the objective. The sequence-level ratio acts as a valid proxy for the global importance weight.

- **Token-Level Invalidity:** In contrast, the token-decomposed ELBO term $\mathcal{L}_\theta^i(y|x)$ is not a valid approximation for the autoregressive conditional probability $\log \pi_\theta(y^i|x, y^{<i})$ required by the Token-level PPO/GRPO formulation (Eq. (16)). The AR conditional $\pi_\theta(y^i|x, y^{<i})$ strictly depends on past tokens $y^{<i}$ (unidirectional), whereas the ELBO term $\mathcal{L}_\theta^i(y|x)$ relies on the bidirectional context of the denoising process. Forcing this mismatched proxy into Eq. (16) introduces an unknown inconsistency, explaining the instability observed in token-level baselines.

In summary, ESPO is a principled algorithm where the intractable likelihood is replaced by a reasonable variational bound (ELBO), constituting the only valid RL formulation for dLLMs under computational constraints.

## C  DETAILS FOR ABLATION STUDY IN SECTION 4

### C.1  ABLATION FOR ACTION SPACE AND LIKELIHOOD APPROXIMATION

To better analyze the objectives used in the ablation study ( Fig. 1), we start by revisiting the GRPO formulation in Eq. (4). For clarity, we focus on its form for a single data sample $(x, y^{(i)})$ of length $L$. This yields the single-sample objective:

$$\mathcal{J}(x, y^{(i)}|\theta) = \frac{\hat{A}^{(i)}}{L} \sum_{k=1}^{L} \min\left(\rho^{k,(i)}, \text{clip}(\rho^{k,(i)}, 1 - \epsilon, 1 + \epsilon)\right), \tag{19}$$

where $\rho^{k,(i)} = \frac{\pi_\theta(y^{k,(i)}|x, y^{<k,(i)})}{\pi_{\theta_{\text{old}}}(y^{k,(i)}|x, y^{<k,(i)})}$.

To highlight the core mechanism, we omit the clipping operator and obtain the simplified form:

$$\mathcal{J}(x, y^{(i)}|\theta) = \frac{\hat{A}^{(i)}}{L} \sum_{k=1}^{L} \rho^{k,(i)}. \tag{20}$$

Building on the simplified form in Eq. (20), we now extend the formulation to dLLMs. Since dLLMs do not provide autoregressive conditional probabilities directly, prior work replaces or approximates $\rho^{k,(i)}$ with diffusion-based likelihood surrogates. This gives rise to four variants used in our ablation:

- **Token-level + Mean-field (Orange Curve):** This approach, introduced by d1 (Zhao et al., 2025), applies a token-wise importance ratio using the mean-field approximation.

$$\mathcal{J}(x, y^{(i)}|\theta) = \frac{\hat{A}^{(i)}}{L} \sum_{k=1}^{L} \frac{p_\theta(y^{k,(i)}|x)}{p_{\theta_{\text{old}}}(y^{k,(i)}|x)} = \frac{\hat{A}^{(i)}}{L} \sum_{k=1}^{L} \exp\left(\log p_\theta(y^{k,(i)}|x) - \log p_{\theta_{\text{old}}}(y^{k,(i)}|x)\right). \tag{21}$$

- **Sequence-level + Mean-field (Green Curve):** This baseline extends the mean-field approach to the sequence level.

$$\mathcal{J}(x, y^{(i)}|\theta) = \hat{A}^{(i)} \cdot \exp\left(\frac{1}{L} \sum_{k=1}^{L} \left[\log p_\theta(y^{k,(i)}|x) - \log p_{\theta_{\text{old}}}(y^{k,(i)}|x)\right]\right). \tag{22}$$

- **Token-level + ELBO (Red Curve):** This method replaces the mean-field term with the token's contribution to the ELBO, but problematically computes the ratio for each token individually. Gong et al. (2025); Yang et al. (2025) follows this approach.

$$\mathcal{J}(x, y^{(i)}|\theta) = \frac{\hat{A}^{(i)}}{L} \sum_{k=1}^{L} \exp\left(\mathcal{L}_\theta^k(y^{(i)}|x) - \mathcal{L}_{\theta_{\text{old}}}^k(y^{(i)}|x)\right). \tag{23}$$

- **Sequence-level + ELBO (Ours, Blue Curve):** Our proposed method. It uses the ELBO as a proxy for the entire sequence log-likelihood and normalizes the log-ratio for stability, as described in Eq. (7) and Eq. (8).

$$\mathcal{J}(x, y^{(i)}|\theta) = \hat{A}^{(i)} \cdot \exp\left(\frac{1}{L}\left[\mathcal{L}_\theta(y^{(i)}|x) - \mathcal{L}_{\theta_{\text{old}}}(y^{(i)}|x)\right]\right)$$
$$= \hat{A}^{(i)} \cdot \exp\left(\frac{1}{L} \sum_{k=1}^{L} \left[\mathcal{L}_\theta^k(y^{(i)}|x) - \mathcal{L}_{\theta_{\text{old}}}^k(y^{(i)}|x)\right]\right). \tag{24}$$

### C.2  ABLATION FOR KL DIVERGENCE ESTIMATOR

The choice of KL-divergence estimator is critical for stable sequence-level optimization. We analyze the three common kl estimators. As demonstrated in Fig. 2 of the main paper, only the $k_2$ estimator enables stable and effective learning. We provide a detailed analysis of their pitfalls here, supported by new visualizations of their training dynamics in Fig. 6.

**Pitfalls in the $k_3$ estimator.** The $k_3$ estimator commonly adopted in GRPO is given by

$$\widehat{\mathbb{KL}}_{k3}(\pi_\theta, \pi_{\text{ref}}) = \mathbb{E}_{y \sim \pi_\theta} \left[ \frac{\pi_\theta(y)}{\pi_{\text{ref}}(y)} - 1 + \log \frac{\pi_\theta(y)}{\pi_{\text{ref}}(y)} \right], \tag{25}$$

where $\pi_\theta$ and $\pi_{\text{ref}}$ are the target and reference policies. However, we argue that the $k_3$ estimator has several pitfalls.

First, the $k_3$ estimator introduces training instability. For a model with the output instead directly parameterizes the $\log$ policy, or logits, the $k_3$ estimator requires to take the exponent:

$$\widehat{\mathbb{KL}}_{k3} = \mathbb{E}_y \left[ \exp \left( \mathcal{L}_\theta(y) - \mathcal{L}_{\text{ref}}(y) \right) - 1 + \mathcal{L}_\theta(y) - \mathcal{L}_{\text{ref}}(y) \right], \tag{26}$$

where $\mathcal{L}(y)$ approximates $\log \pi(y)$. We empirically find that during RL training $\mathcal{L}(y)$ may incur instability which will be magnified by the exponent, *i.e.*, the first term in Eq. 26. As shown in Fig. 6(right column), the $k_3$ estimate and its gradient norm are characterized by extreme, infrequent spikes, which destabilize and derail the entire training process.

Besides, while the $k_3$ estimator itself is an unbiased estimate of KL-divergence, its gradient is not (Tang & Munos, 2025a). In fact, it is shown that its gradient is instead an unbiased estimate of reverse-KL (Tang & Munos, 2025a):

$$\mathbb{E} \left[ \widehat{\nabla \mathbb{KL}}_{k3}(\pi_\theta, \pi_{\text{ref}}) \right] = \mathbb{E} \left[ \widehat{\nabla \mathbb{KL}}(\pi_{\text{ref}}, \pi_\theta) \right] \neq \mathbb{E} \left[ \widehat{\nabla \mathbb{KL}}(\pi_\theta, \pi_{\text{ref}}) \right]. \tag{27}$$

Therefore, the optimization using $k_3$ estimator will exploit gradients that indeed optimize towards the reverse-KL, which will empirically lead to a different converged policy with bounded model capacity.

**Pitfalls in the $k_1$ estimator.** Another alternative is the original $k_1$ estimator for KL-divergence, which is given by

$$\widehat{\mathbb{KL}}_{k1}(\pi_\theta, \pi_{\text{ref}}) = \mathbb{E}_{y \sim \pi_\theta} \left[ \log \frac{\pi_\theta(y)}{\pi_{\text{ref}}(y)} \right]. \tag{28}$$

However, by taking the gradient *w.r.t.* $\theta$ on Eq. 28, it is proved(Tang & Munos, 2025a) that the gradient equals zero, which means the $k_1$estimator has no signal of the KL constraint. This is empirically confirmed in Fig. 2 (right) of the main paper, where the $k_1$ policy (orange) initially learns but then catastrophically collapses. Fig. 6 (left column) further illustrates this: the KL estimate plummets into negative values, and the gradient norm grows uncontrollably, reflecting a complete loss of stability as the policy drifts unconstrained.

**The $k_2$ estimator.** We propose to employ the $k_2$ estimator

$$\widehat{\mathbb{KL}}_{k2}(\pi_\theta, \pi_{\text{ref}}) = \mathbb{E}_{y \sim \pi_\theta} \left[ \frac{1}{2} \left( \log \pi_\theta(y) - \log \pi_{\text{ref}}(y) \right)^2 \right], \tag{29}$$

which leads to

$$\widehat{\mathbb{KL}}_{k2} = \mathbb{E}_{y \sim \pi_\theta} \left[ \frac{1}{2} \left( \mathcal{L}_\theta(y) - \mathcal{L}_{\text{ref}}(y) \right)^2 \right], \tag{30}$$

under the $\log$ parameterization. The $k_2$ estimator takes the form of an MSE loss that does not involve an exponential term. As shown in Fig. 6 (middle column), both the KL estimate and its gradient norm are exceptionally stable, smooth, and well-behaved throughout training. Furthermore, the gradient of the $k_2$ estimator is indeed unbiased (Tang & Munos, 2025a):

$$\mathbb{E} \left[ \widehat{\nabla \mathbb{KL}}_{k2}(\pi_\theta, \pi_{\text{ref}}) \right] = \mathbb{E} \left[ \widehat{\nabla \mathbb{KL}}(\pi_\theta, \pi_{\text{ref}}) \right] \tag{31}$$

as opposed to the biased estimate in $k_3$ estimator.

This analysis, confirms that $k_2$ is the only robust and theoretically sound choice for our high-variance, sequence-level optimization framework.

## D  VARIANCE REDUCTION

As discussed in Section 2.1, there are two equivalent formulations of the ELBO. Empirically, Eq. (3) exhibits lower variance than Eq. (2) (Ou et al., 2024; Nie et al., 2025). Therefore, we adopt Eq. (3) as our estimator for the ELBO. To further stabilize training, we incorporate two variance reduction techniques: antithetic sampling through mask sharing (Zhu et al., 2025) and coupled sampling (Gong et al., 2025), which we describe below.

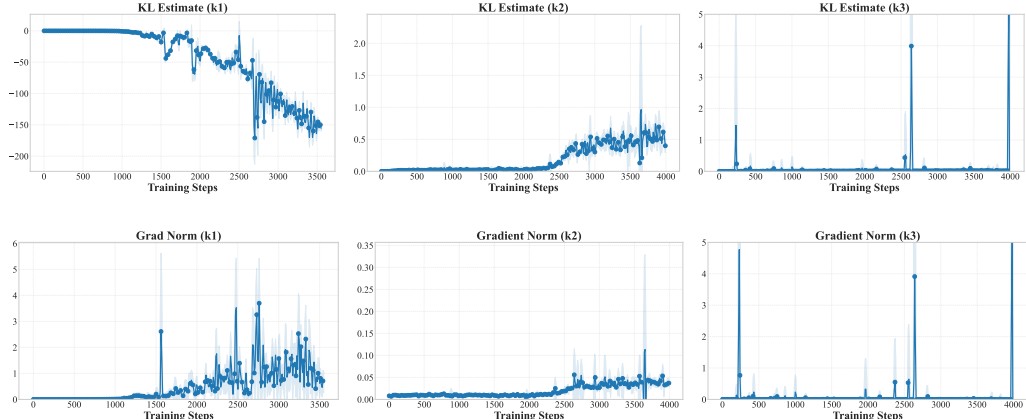

Figure 6: **Comparison of training dynamics with different KL divergence estimators.** The top row illustrates the KL estimates, while the bottom row displays the gradient norms for the $k_1$, $k_2$, and $k_3$ estimators. The $k_1$ estimator (left) lacks effective constraints, leading to model collapse. The $k_3$ estimator (right) suffers from severe instability and gradient spikes. In contrast, the $k_2$ estimator (middle) demonstrates superior stability and consistent gradient norms throughout training.

**Antithetic Sampling through Mask Sharing.** Antithetic sampling through mask sharing is used when computing the difference between two ELBO terms, such as $\mathcal{L}_\theta(y|x) - \mathcal{L}_{\theta_{\mathrm{old}}}(y|x)$ in Eq. (8) and $\mathcal{L}_\theta(y|x) - \mathcal{L}_{\theta_{\mathrm{ref}}}(y|x)$ in Eq. (10). Concretely, we share the sampled timesteps and masked positions between the Monte Carlo estimators of the two policies, thereby reducing variance through negative correlation.

**Coupled Sampling.** Coupled sampling was originally proposed for the ELBO in the form of Eq. (2) (Gong et al., 2025), but we adapt it to Eq. (3). Note that Eq. (3) is equivalent to

$$\mathbb{E}_{l \sim \mathcal{U}(\{0,1,2,\ldots,L\})}\mathbb{E}_{y_l \sim q(y_l|l,y,x)}[\ell_\theta(y_l, l, y|x)], \tag{32}$$

where

$$\ell_\theta(y_l, l, y|x) \triangleq \begin{cases} \dfrac{L+1}{l} \displaystyle\sum_{i=1}^{L} \mathbf{1}[y_l^i = \mathbf{M}] \log p_\theta(y^i \mid y_l, x), & l > 0, \\ 0, & l = 0. \end{cases} \tag{33}$$

Based on this formulation, we introduce a complementary masking strategy. For each sampled mask $y_l$, we construct a complementary mask $\tilde{y}_l$ such that the two masks partition the token set: every token masked in $y_l$ is unmasked in $\tilde{y}_l$, and vice versa. We then average the two losses, which is also equivalent:

$$\mathbb{E}_{l \sim \mathcal{U}(\{0,1,2,\ldots,L\})}\mathbb{E}_{y_l \sim q(y_l|l,y,x)}\left[\frac{\ell_\theta(y_l, l, y|x) + \ell_\theta(\tilde{y}_l, L-l, y|x)}{2}\right]. \tag{34}$$

This construction guarantees that every token contributes at least once to the learning signal. Further, the estimator achieves lower variance and yields a more stable optimization objective.

# E EXPERIMENT DETAILS

As described in Section 5.1, we train separate models for each of the GSM8K, Math, Sudoku, and Countdown tasks using their respective training datasets. For code tasks, we train a unified model on AceCoder-87K (Zeng et al., 2025) and evaluate it on four benchmarks: HumanEval, HumanEval-Plus, MBPP, and MBPP-Plus. For ablation experiments, we conducted only on LLaDA-Instruct-8B model.

All models are trained with a maximum sequence length of 256, while evaluation is performed at sequence lengths of 128, 256, and 512 to better assess length generalization.

## E.1 Inference Setting

For simpler planning tasks (Sudoku and Countdown), we unmask 2 tokens per denoising step, resulting in $L/2$ total denoising steps (where $L = 256$ is the sequence length). For other tasks, the number of denoising steps is set equal to the sequence length to improve performance.

Sampling strategies are model-specific:

- **LLaDA**: We employ low-confidence sampling (Chang et al., 2022) combined with a semi-autoregressive decoding strategy (Arriola et al., 2025; Nie et al., 2025), with block length set to 32 for all tasks. Training and evaluation share most sampling settings, except that the temperature is 0.9 during training and 0 (greedy decoding) during evaluation, following the evaluation codebase (Nie et al., 2025).
- **Dream**: We use top negative entropy remasking and pure diffusion sampling, without semi-autoregressive decoding (Ye et al., 2025). Temperature is set to 0.9 during training and 0.1 during evaluation, while other sampling settings remain consistent.

## E.2 Training Details

All reinforcement learning training is conducted using the TRL library (von Werra et al., 2020).

**Parameter-Efficient Fine-Tuning** For GSM8K, Math, Countdown, and Sudoku, we apply LoRA (Hu et al., 2022) with rank $r = 128$ and scaling factor $\alpha = 64$. For code tasks, full parameter fine-tuning is used to maximize performance.

**Optimization** Policy update value is set to $\mu = 8$, and the number of Monte Carlo (MC) samples is $M = 2$ for computational efficiency. Models are optimized using AdamW (Loshchilov & Hutter, 2019) with $\beta_1 = 0.9$, $\beta_2 = 0.99$. A constant learning rate schedule is used. For LoRA-based tasks, the learning rate is $3 \cdot 10^{-6}$ with weight decay 0.01 and gradient clipping 0.2. For code tasks, the learning rate is $1 \cdot 10^{-6}$, weight decay 0.1, and gradient clipping 0.8.

**Batching** We set the group size $G$ in Eq. (7) and total batch size according to the difficulty of the tasks: GSM8K, Countdown, Sudoku use $G = 6$ and total batch size 96, Math uses $G = 16$ and batch size 256, and code tasks use $G = 10$ and batch size 160. Gradient accumulation is applied to enlarge the effective batch size.

**Training Steps** Models are evaluated when the reward curve converges. Code tasks are trained for 2k steps, Math for 3k steps, LLaDA planning tasks for 10k steps, and Dream planning tasks for 8k steps.

**Special Notes on Sudoku** The Sudoku setup in Dream (Ye et al., 2025) differs from d1 (Zhao et al., 2025). The prompt and data format in Dream are inconsistent with d1 (see Appendix H.1). Additionally, Dream restricts the generation length to 24 tokens, encouraging direct answers without intermediate reasoning. In contrast, d1 uses longer generation lengths (128/256/512), allowing the model to produce both intermediate reasoning steps and final answers.

In the main text, we report results under the d1 setting, as it better reflects the ability of models to perform multi-step reasoning when guided by our reinforcement learning algorithm. For completeness, we also present model performance under the Dream setting in Table 4.

**Special Notes on Coding** It is important to note that for coding tasks, we employed full fine-tuning, unlike the LoRA setting used for Math and Planning. In this regime, we observed significant instability with the baseline methods. Specifically, wd1 shows to be highly sensitive to hyperparameters; using default settings resulted in exploding gradient norms (reaching $10^3 \sim 10^4$) and immediate training collapse. While we were able to stabilize wd1 by carefully tuning the hyperparameters, it still suffered from performance degradation on certain metrics compared to the base model, as shown in Table 2. In contrast, ESPO remained stable and effective.

Table 4: **Model performance on Sudoku under the Dream setting (generation length = 24).** Unlike the d1 setting, the Dream setting restricts generation to 24 tokens, encouraging models to output answers directly without intermediate reasoning.

| Model | Accuracy (%) |
|---|---|
| Dream-7B-Instruct | 98.1 |
| + diffu-GRPO (d1) | 96.0 |
| + ESPO (ours) | 98.0 |

## F    DETAILS OF TRAINING FLOPS

Recent advances in dLLMs (Wu et al., 2025; Ma et al., 2025; Liu et al., 2025c) enable partial usage of KV cache, but it will lead to the loss of downstream performance. Therefore, our analysis focuses on the vanilla case without KV cache.

As we mentioned in Section 5.5, for a model with $N$ parameters and sequence length $D$, we use $C_{\text{forward}} = 2ND$ and $C_{\text{backward}} = 4ND$ to approximate the forward and backward passes FLOPS according to Kaplan et al. (2020). Combining the costs of generation and policy updates, we obtain the naive formula of total FLOPs per sample:

$$F_{\text{total}} = K \cdot C_{\text{forward}} + \mu M(C_{\text{forward}} + C_{\text{backward}}) = 2ND(K + 3\mu M), \tag{35}$$

where $K$ is the sampling step, $\mu$ is the policy update values, and $M$ is the number of MC samples. When using the coupled-sampling technique ,we will double the FLOPS of policy updates step, which leads to:

$$F_{\text{total}} = K \cdot C_{\text{forward}} + 2\mu M(C_{\text{forward}} + C_{\text{backward}}) = 2ND(K + 6\mu M), \tag{36}$$

Which indicates increasing $M$ only adds to the policy update term ($6\mu M$ in Eq. (36)). When $K \gg 6\mu M$, the overall training cost grows mildly with $M$.

## G    ADDITIONAL EXPERIMENTAL RESULTS

### G.1    DETAILED TRAINING DYNAMICS OF SUDOKU

As shown in Fig. 7, the training dynamics exhibit a clear phase transition characterized by a sharp rise in reward. During the initial stage, the model explores with low rewards and stable gradients. However, coinciding exactly with the sudden performance jump (around step 2500), we observe a sharp spike in the gradient norm and a rapid surge in KL divergence. These signals indicate that the model has successfully discovered the strict logical rules of Sudoku, triggering a rapid restructuring of its policy and a decisive shift away from the reference model to exploit the newly discovered solution mechanism.

### G.2    TRAINING CURVES ON MATHEMATICAL BENCHMARKS

As illustrated in Fig. 8, the training reward trends for ESPO and the baselines are relatively stable and similar, indicating that all evaluated methods can robustly fit the training distribution given the reward signals. However, as observed in prior work (Liu et al., 2025d), the final reward on the training set often exhibits a weak correlation with the accuracy on the held-out validation set due to the potential for overfitting. While the training rewards are similar, ESPO's superior performance on the test set (as reported in Table 1) demonstrates that our sequence-level optimization leads to better generalization rather than mere memorization of the training data.

### G.3    ABLATION STUDY ON KL ESTIMATOR WITH TOKEN-LEVEL BASELINES

To rigorously investigate whether the superior performance of ESPO is primarily driven by the robust $k_2$ KL estimator rather than our sequence-level framework, we conducted an additional ablation study on the token-level baseline. The wd1 method (Tang et al., 2025) reformulates the original RL

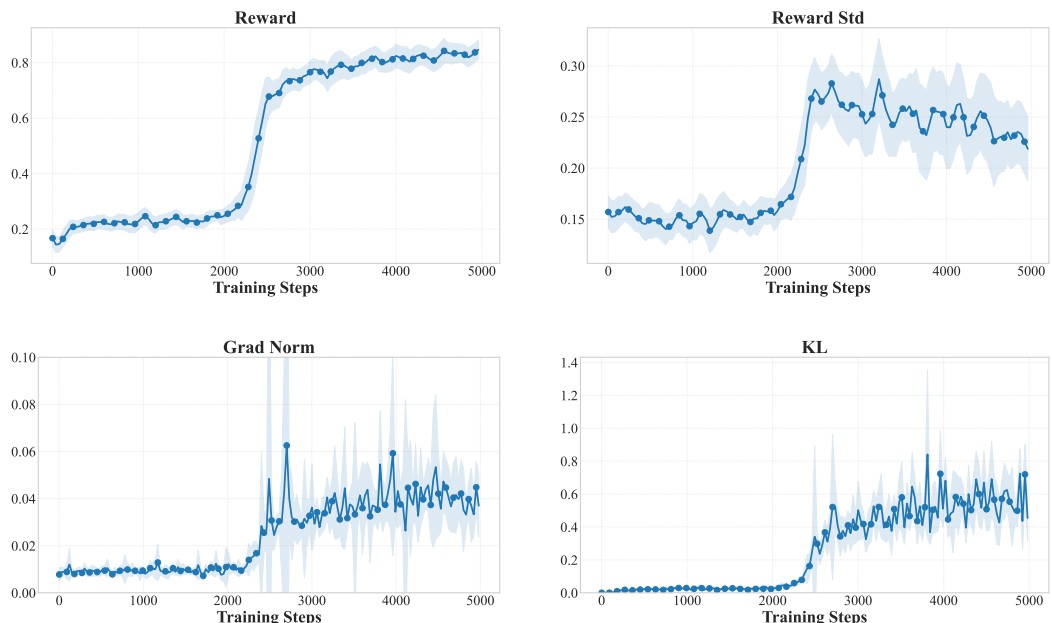

Figure 7: **Detailed training dynamics of the Sudoku Task on LLaDA-8B-Instruct.** We track reward, reward standard deviation, gradient norm, and KL divergence to analyze the learning process of our ESPO method.

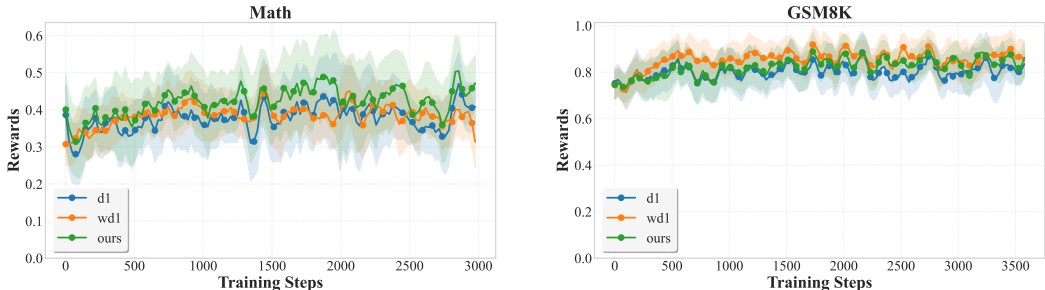

Figure 8: **Training dynamics with different methods for MATH and GSM8K benchmarks on LLaDA-8B-Instruct.**

objective as a weighted likelihood objective (similar to AWR) and does not involve an explicit KL estimation. Therefore, we only conducted ablations for diffu-GRPO (d1)Zhao et al. (2025) in the sudoku task, replacing its original unstable $k_3$ estimator with the robust $k_2$ estimator in Table 5.

It shows that simply applying the $k_2$ estimator to the token-level baseline (d1 + $k_2$) only yields performance comparable to the original baseline (d1 + $k_3$). Both variants stagnate at a low success rate ($\sim 20 - 26\%$). In stark contrast, ESPO achieves a dramatic improvement ($> 80\%$).

Table 5: **Performance comparison on Sudoku (LLaDA-8B-Instruct).** We compare the original d1 (using $k_3$), d1 adapted with the $k_2$ estimator, and our ESPO method.

| Method / Seq Len | 128 | 256 | 512 |
|---|---|---|---|
| diffu-GRPO (d1) + $k_3$ (Original) | 26.7 | 24.1 | 15.9 |
| diffu-GRPO (d1) + $k_2$ (Ablation) | 26.3 | 23.1 | 18.8 |
| **ESPO (Ours) + $k_2$** | **92.7** | **84.7** | **80.5** |

## H  PROMPTS AND OUTPUTS

### H.1  PROMPT TEMPLATES FOR SUDOKU

Below, we demonstrate the one-shot prompts used for the Sudoku task.

---

**One-shot Prompt (d1-style)**

Please solve the following 4x4 Sudoku puzzle. The puzzle is provided as a 16-character string reading left-to-right, top-to-bottom, where '0' represents empty cells.
**Rules:**

- Fill empty cells with digits 1-4.
- Each row must contain digits 1-4 exactly once.
- Each column must contain digits 1-4 exactly once.
- Each 2x2 box must contain digits 1-4 exactly once.

**Example:**
Puzzle: `0401002010030310`
This puzzle grid looks like this:

```
0 4 | 0 1
0 0 | 2 0
----+----
1 0 | 0 3
0 3 | 1 0
```

Solution: `2431312412434312`
The solved grid looks like this:

```
2 4 | 3 1
3 1 | 2 4
----+----
1 2 | 4 3
4 3 | 1 2
```

**Important:** Your solution must be a COMPLETE 16-character string with only the digits 1-4, representing your final solved grid.
Respond in this exact format:
```
<reasoning>
```
Your step-by-step solving process
```
</reasoning>
<answer>
```
[16-character solution string]
```
</answer>
```

Now, solve the following Sudoku puzzle: `1240300020140100`

---

**One-shot Prompt (Dream-style)**

```
<|im_start|>system
You are an AI assistant skilled at solving Sudoku puzzles.<|im_end|>
<|im_start|>user
```
Fill the positions where the values are 0 in a 4x4 grid with digits 1-4 so that each column, each row, and each of the four 2x2 subgrids that compose the grid contains all of the digits from 1 to 4.
Input:

```
0321
0030
0400
2100
```

Output:

```
4321
1234
3412
2143
```

Input:

```
0001
2100
1430
3000
```

Output:
```
<|im_end|>
<|im_start|>assistant
```

## H.2   QUALITATIVE EXAMPLES ON SUDOKU TASK

**🤖 LLaDA + d1 (Baseline)**

```
<reasoning>
```
To solve this 4x4 Sudoku puzzle, we need to fill in the empty cells with digits 1-4 such that each row, column, and 2x2 box contains all digits exactly once. We can start by examining the given digits and trying to fill them in according to the Sudoku rules.

1. The first row has the 0' and the 4's in the first and fourth columns.
2. The second row has the 4's in the first and fourth columns.
3. The third row has the 3 in the second column.
4. The fourth row has the 2 in the fourth column.
5. The fifth row has the 1 in the first column.
6. The fourth row has the 3 in the first column.
7. The first row has the 4 in the fourth column.

Using this information, we can start filling in the grid. We need to ensure that each row, column, and 2x2 box contains the digits 1-4 exactly once.
```
</reasoning>
<answer>
12340
40221
30114
</answer>
```

---

**★ LLaDA + ESPO (Ours)**

```
<reasoning>
```
To solve the 4x4 Sudoku puzzle, we need to fill in the grid with digits 1-4 such that each row, column, and 2x2 box contains all the digits exactly once. We can start by filling in the cells with the given digits and check for conflicts.

First, let's fill in the grid with the given digits:
3 2 0 4
0 0 3 0
1 4 0 0
2 0 0 1

Next, let's try filling fill in the remaining with digits 1-4, ensuring each row, column, and 2x2 box contains all the digits once.

After trying different combinations, we find a valid solution:
3 2 1 4
4 1 3 2
1 4 2 3
2 3 4 1

This solution satisfies all the rules: each row, column, and box contains all digits 1-4 exactly once.

```
</reasoning>
```

```
<answer>
3214413214232341
</answer>
```

---

## H.3 QUALITATIVE EXAMPLES ON COUNTDOWN TASK

**LLaDA+d1 (Baseline)** — Target: 94 — Numbers: [77, 73, 98]

```
<reasoning>
```
To create an arithmetic expression that evaluates to 94 using the numbers [77, 73, 98], we need to find the correct combination of operations (+, -, *, /) that will inbalance the the larger number 98 and the smaller numbers to 94.
```
</reasoning>
<answer>
```
77*73-98
```
</answer>
```
✗*(Incorrect, evaluates to 5523)*

**LLaDA+ESPO (Ours)** — Target: 67 — Numbers: [72, 92, 47]

```
<reasoning>
```
To create an arithmetic expression that evaluates to exactly 67 using the numbers [72, 92, 47], we need to consider the operations +, -, *, and / Let's start by examining the largest numbers and see if they can combine to give us target 67.

1. If we add 92 and 72, we get 164, which is too than 67.
2. If we subtract 92 and 72, we get 20, which is less than 67.
3. If we add 92 and 47, we get 139, which is more than 67.
4. If we subtract 72 and 47, we get 25, which is less than 67.

. If we combine 92 and 72, we get 164, which is more than 67.

Combining 92 and 47 will the closest answer to 67.
```
</reasoning>
<answer>
```
92 -72 + 47
```
</answer>
```
✓*(Correct, $20 + 47 = 67$)*

