# OpenReview forum: "Principled RL for Diffusion LLMs Emerges from a Sequence-Level Perspective"
_ICLR.cc/2026/Conference — ICLR 2026 Poster_

### Official Review · Reviewer_b9P2 · 2025-10-29

**Soundness:** 4
**Presentation:** 4
**Contribution:** 2
**Rating:** 4
**Confidence:** 3

**Summary:**

This work proposes ELBO-based Sequence-level Policy Optimization (ESPO), a principled framework tailored for dLLMs. ESPO defines the entire sequence generation as an atomic action (avoiding token-level decomposition), employs the evidence lower bound (ELBO) as a tractable proxy for sequence-level likelihood, and utilizes a robust $k_2$ estimator for KL divergence (free from exponential instability).

Extensive experiments on models like LLaDA-8B-Instruct and Dream-7B-Instruct across tasks (mathematical reasoning, coding, planning) demonstrate ESPO’s superiority: it achieves up to 60-point absolute improvements in planning tasks (e.g., Sudoku), stable gains in knowledge-intensive tasks (e.g., GSM8K, HumanEval), and superior training efficiency (only 47% cost increase when MC samples double).

**Strengths:**

1. Clearly identifies the mismatch issue under the dLLM framework, highlighting that existing RL algorithms struggle to compute token-level importance sampling, and proposes a sequence-level alternative.

2. The experiments are thorough, including extensive validation and ablation studies across both math reasoning and agent tasks, demonstrating the effectiveness of the proposed method.

3. The paper is well-organized, with clear presentation and coherent argumentation.

**Weaknesses:**

1. For the math reasoning experiments, could the authors provide training curves? The improvement over the baseline appears limited.

2. Why are there no experiments on Dream-7B-Instruct with the d1 baseline?

3. Why are the results on coding benchmark such as HumanEval and MBPP missing baseline comparisons?

4. The performance on Sudoku shows a sudden change. Could the authors provide a deeper explanation or analysis?

5. From Figure 2, the observed improvements seem mainly driven by $k_2$ estimator. Could the authors include wd1 and d1 with $k_2$ estimator on Sudoku as additional comparisons?

I am willing to raise my score should the authors provide satisfactory responses and clarifications.

**Questions:**

Please see questions in Weaknesses.

---

> ### Author Response · Authors · 2025-11-21
> **Response to Reviewer b9P2**
>
> Thank you for your constructive feedback on our paper. Below, we address your concerns and suggestions.
>
>
> ### **W1: For the math reasoning experiments, could the authors provide training curves?**
>
> We have included the training curves for MATH and GSM8K in Figure 8. As shown in the figures, the training rewards are similar across different methods. You can refer to the updated Appendix G.2 for a detailed discussion on the training dynamics and its relation to final performance.
>
> ### **W2: Missing experiments on Dream-7B-Instruct with the d1 baseline.**
>
> We respectfully clarify that the results for Dream-7B-Instruct + d1 were actually already present in Table 1 of our original submission. However, assuming you intended to inquire about the wd1 baseline (which was missing for Dream), we have conducted these additional experiments. The comparison between wd1 and ESPO on Dream-7B-Instruct is summarized below. We report the average performance across all generated lengths (128, 256, and 512) below and update the full results in Table 1 of the revised paper. ESPO consistently outperforms wd1 on the Dream model as well.
>
> |Method|GSM8K|MATH(0)|Countdown|Sudoku|
> |--|--|--|--|--|
> |Dream-7B + wd1 | 80.5 |45.8|36.3|33.0|
> |Dream-7B + ESPO (Ours)|81.3|46.0|66.8 |71.8|
>
>
>
> ### **W3: Why are the results on coding benchmark such as HumanEval and MBPP missing baseline comparisons?**
>
> Thank you for pointing this out. We initially prioritized planning tasks due to compute constraints, but we have now completed the baseline comparisons for coding tasks. The table below summarizes the performance of LLaDA-8B-Instruct on HumanEval, HumanEval+, MBPP, and MBPP+. We report the average performance across all generated lengths (128, 256, and 512)  below and update the full results in Table 2 of the revised paper.
>
> | Method | HumanEval | HumanEval+ | MBPP | MBPP+ |
> | :--- | :--- | :--- | :--- | :--- |
> | LLaDA-8B + d1 (diffu-GRPO) | 37.2 | 29.7 | 36.5 | 35.3 |
> | LLaDA-8B + wd1 | 37.2 | 30.9 | 36.5 | 33.4 |
> | **LLaDA-8B + ESPO (Ours)** | **40.1** | **34.6** | **45.4** | **41.0** |
>
> As shown, ESPO consistently outperforms both d1 and wd1, particularly on MBPP (+8.9%). It is important to note that for coding tasks, we employed full fine-tuning, unlike the LoRA setting used for Math and Planning. In this regime, we observed significant instability on the baseline methods. We have added a detailed discussion in Appendix E of the revised paper.
>
>
> ### **W4: The performance on Sudoku shows a sudden change. Could the authors provide a deeper explanation or analysis?**
>
> Thank you for pointing this out. We have added a detailed analysis in Appendix G.1, including additional training metrics such as gradient norm and KL divergence in Figure 7. As it shows, the performance jump coincides exactly with a distinctive spike in gradient norm and a rapid rise in KL divergence. Based on these signals, we believe the jump represents a genuine learning milestone where the model successfully grounds the Sudoku rules and significantly updates its policy distribution.
>
>
>
> ### **W5: Comparison with d1/wd1 + $k_2$.**
>
> This is a critical question. We have added additional ablations in Appendix G.3 of our revised paper to show that the improvements are not solely due to the use of the $k_2$ estimator.
>
> - Clarification on wd1: The **wd1** method (Tang et al., 2025) reformulates the original RL objective as a weighted likelihood objective (similar to AWR) and does not involve an explicit KL estimation. Therefore, the choice of KL estimator is not applicable to wd1.
>
> - Comparison with d1 + $k_2$: To address your concern regarding d1, we trained the **d1** (diffu-GRPO) baseline using the robust $k_2$ estimator on the Sudoku task. The results are compared below:
>
> | Method | Length 128 | Length 256 | Length 512 |
> | :--- | :--- | :--- | :--- |
> | d1 + $k_3$ (Original) | 26.7 | 24.1 | 15.9 |
> | **d1 + $k_2$ (Ablation)** | **26.3** | **23.1** | **18.8** |
> | **ESPO (Ours)** | **92.7** | **84.7** | **80.5** |
>
>
> This experiment serves as strong evidence that the "observed improvements" are not driven by the $k_2$ estimator alone.

---

> > ### Comment · Reviewer_b9P2 · 2025-11-23
> >
> > Thank you for your answer, it solved my problem. I’ve increased rating to 6.

---

> > > ### Author Response · Authors · 2025-11-24
> > >
> > > Thank you for your time and acknowledgment of our contributions. We are glad that your concerns have been addressed, and your constructive suggestions have improved the quality of our paper.

---

### Official Review · Reviewer_sYDr · 2025-10-30

**Soundness:** 2
**Presentation:** 3
**Contribution:** 1
**Rating:** 4
**Confidence:** 4

**Summary:**

This paper proposes a reinforcement learning framework for training diffusion language models. The approach builds upon group sequence-level importance sampling ratios for diffusion language models and incorporates an additional KL regularization term (referred to as K2-type regularization by the authors). The method is evaluated empirically on several tasks: Countdown, mathematical reasoning, and coding problems, demonstrating performance improvements over baseline approaches.

**Strengths:**

- The paper is well-written with a logical structure that makes the technical content accessible. The progression from problem formulation to methodology to experimental validation is easy to follow.

- The experimental evaluation demonstrates notable improvements on both the Countdown and math coding tasks, suggesting the proposed approach is effective for the target applications.

**Weaknesses:**

-  The proposed method largely combines existing techniques from prior work (Zheng et al., 2025; Tang & Munos, 2025b) without introducing new algorithmic components or theoretical insights. The contribution appears primarily incremental, adapting established methods to the diffusion language model setting rather than developing new approaches tailored to the unique characteristics of these models.
- While the empirical improvements are encouraging, the paper does not address the fundamental question of what constitutes an appropriate RL formulation for diffusion language models. The current approach treats the problem as one of better approximating log-probabilities, but a more principled direction would be to re-derive the policy gradient theorem specifically for diffusion language models from first principles. Critical questions remain unanswered: What is the proper form of policy gradients for diffusion language models? Does the "log-probability" term even appear in such gradients, or should the formulation take a fundamentally different form? Without addressing these foundational issues, the work risks building on potentially unsuitable assumptions.

**Questions:**

**Justification for sequence-level formulation:** The paper does not provide clear insight into why the proposed sequence-level formulation outperforms the token-level formulation. Is the advantage due to reduced bias, reduced variance in the policy gradient estimates, or both? A quantitative analysis comparing the bias-variance tradeoffs of both formulations would strengthen the paper's claims.

**Hyperparameter selection:** How was the clipping parameter $\epsilon$ in Proximal Policy Optimization chosen? Were separate hyperparameter searches conducted for the token-level and sequence-level formulations, or was the same value used for both? This is important for ensuring fair comparison between the two approaches.

**Role of KL regularization:** The results in Figure 2 raise several questions about KL regularization:

The figure suggests that KL regularization enables a significant performance breakthrough in later training stages, which is unexpected. Typically, KL regularization is understood to improve training stability rather than drive performance gains. Can the authors explain this phenomenon?
- In standard language model fine-tuning on reasoning tasks, KL regularization is often minimal or omitted entirely. Why is it critical in this setting?
- For clarity: which type of KL regularization (forward, reverse, or the K2-type mentioned) is used in Figure 1?

---

> ### Author Response · Authors · 2025-11-21
> **Response to Reviewer sYDr**
>
> Thank you for the detailed review. We have tailored our rebuttal to address the points you raised.
>
>
> ### **W & Q1: Theoretical Grounding & Novelty for sequence-level formulation**
> We thank the reviewer for the insightful and rigorous feedback. We particularly appreciate the challenge regarding the "first principles" derivation, which prompted us to formalize the theoretical foundations of our method in the newly added Appendix B.
>
> In this analysis, we emphasize that the exact RL formulation for dLLMs is trajectory-level optimization. However, this approach is computationally prohibitive for large-scale training, as it requires backpropagating through the entire denoising chain ($T$ steps). This necessitates a computationally efficient method to approximate the log-likelihood.
>
> From this perspective, we analyze the advantages and disadvantages of both token-level and sequence-level approaches. We find that token-level methods face fundamental computational obstacles in dLLMs, whereas sequence-level methods can efficiently approximate the log-likelihood and thereby enable more effective RL training. This makes ESPO the framework that achieves a better balance between theoretical validity and computational feasibility. More detailed mathematical derivations can be referenced in Appendix B.
>
>
>
>
>
>
>
>
>
>
>
>
>
>
>
>
> ### **Q2: Hyperparameter selection**
>
> To maintain a fair comparison, we used the exact same PPO clipping parameter $\epsilon=0.2$ for all methods, which is used in prior works such as d1 and wd1.
>
> ### **Q3: Role of KL regularization**
>
> To clarify, we employed the $k_2$ estimators as KL regularization in Figure 1.
>
> The pronounced sensitivity of dLLMs to KL regularization arises from the nature of sequence-level policy updates, where we compute the probability of entire sequences. This leads to a broader numerical range and more severe numerical challenges in KL estimation compared to token-level approaches.
>
> In prior work on sequence-level RL for AR  (Zheng et al., 2025), KL regularization was removed; however, we found that training dLLMs without KL regularization resulted in significant instability and possible training collapses. We hypothesize that this instability stems from the need for dLLMs to approximate the exact likelihood using ELBO. In the updated Appendix C.2 of our paper, we have included training curves for KL divergence and gradient norms using $k_1$, $k_2$, and $k_3$ estimators, along with a detailed analysis.

---

### Official Review · Reviewer_8CJA · 2025-10-31

**Soundness:** 3
**Presentation:** 2
**Contribution:** 3
**Rating:** 6
**Confidence:** 3

**Summary:**

The paper proposes ESPO, an RL framework tailored to diffusion LLMs which treats the whole completion as a single action. Experiments on LLaDA-8B-Instruct and Dream-7B-Instruct across math (GSM8K, MATH), coding (HumanEval/MBPP and EvalPlus), and planning (Countdown, Sudoku) show consistent gains versus token-level d1/wd1 baselines.

**Strengths:**

- Using a sequence-level action space makes the method very simple
- Experiments show strong improvements over the baselines which do not treat the entire sequence as an action

**Weaknesses:**

- Seems like a straightforward application of GSPO to diffusion models novelty-wise
- Not clear why per-token evaluation is necessarily bad for diffusion LLMs specifically - is it true for all LLMs (as GSPO claims) or just diffusion LLMs?

**Questions:**

See weaknesses

---

> ### Author Response · Authors · 2025-11-21
> **Response to Reviewer 8CJA**
>
> We thank the reviewer for recognizing the simplicity and strong performance of our method. We would like to clarify why sequence-level optimization is uniquely critical for dLLMs.
>
> ### **W & Q: Novelty compared to GSPO & Why token-level evaluation is specifically problematic for dLLMs.**
>
>
> Sequence-level RL is generally beneficial for all LLMs , but it brings additional, fundamentally more important advantages for dLLMs.
>
> On the one hand, in both AR models and dLLMs, mainstream RL reward designs grant rewards at the sequence level(output reward), and applying off-policy correction at the token level is inherently improper. On the other hand, token-level RL requires token-level conditional probabilities that are easy to compute for AR models but intractable for dLLMs. For dLLMs, the most reasonable and tractable approximation to likelihood is the ELBO which is defined at sequence level.Existing token-level dLLM baselines rely on mathematically strateforward approximations (e.g., mean-field or splitting sequence-level ELBO), which we show leads to performance degradation (Figure 1). You can reference our detailed theoretical analysis in Appendix B of the revised paper.
>
> Empirically, for AR models, the main benefit of GSPO arises from stabilizing RL training for large MoE architectures, while its performance on dense models is similar to GRPO [1*]. In contrast, for dLLMs, even on dense models such as LLaDA and Dream, our sequence-level RL objective significantly outperforms all token-level baselines. These theoretical and empirical observations demonstrate that the structural properties of dLLMs make sequence-level RL not only beneficial but necessary. We will add this discussion in the camera-ready version.
>
>
>
> [1*] Official Discussion on GSPO performance by the first author of GSPO: https://github.com/volcengine/verl/pull/2775#issuecomment-3134375131

---

### Official Review · Reviewer_Twof · 2025-11-01

**Soundness:** 3
**Presentation:** 3
**Contribution:** 3
**Rating:** 8
**Confidence:** 3

**Summary:**

The paper studies the problem of reinforcement learning (RL) to diffusion LLMs. Token-level objectives for autoregressive model (e.g., GRPO) require token log-likelihoods that are intractable to compute for diffusion models. The paper proposes ESPO, a sequence-level policy optimization method that treats generating the entire completion as one action and replaces the intractable sequence log-likelihood with an ELBO proxy. ESPO stabilizes training by (i) normalizing the ELBO log-ratio by sequence length and (ii) using a k2 (quadratic) KL estimator instead of the unstable exponential k3 estimator, alongside variance-reduction tricks (antithetic/coupled masking). ESPO is empirically evaluated with LLaDA-8B-Instruct and Dream-7B-Instruct on math (GSM8K, MATH), coding (HumanEval/MBPP), and planning (Countdown, Sudoku). ESPO consistently beats diffu-GRPO and wd1. Ablations show sequence-level+ELBO is the only stable formulation among tested variants and a training-cost analysis shows FLOPs/time grow mildly with Monte-Carlo samples because generation dominates compute.

**Strengths:**

* The paper explains the problem setup and why token-level importance ratios lack a valid probabilistic interpretation for dLLMs fairly well. The shortcomings of existing methods also makes the motivations quite clear.

* I like the idea of moving to a sequence level objective and using an ELBO-based ratio avoiding heuristic token surrogates.

* The performance of the method is tested across a variety of tasks and two different base models and shows consistent improvement over the baselines.

* The paper is also generally quite well written.

**Weaknesses:**

* ESPO optimizes an ELBO difference, not the true sequence likelihood ratio but the paper does not quantify how ELBO tightness affects policy improvement or bias across tasks.
* The paper misses some closely related prior work on RL fine-tuning of diffusion language models [1, 2]. I believe a comparison to these baselines would be critical.

[1] Venkatraman et al., 2024. Amortizing intractable inference in diffusion models for vision, language, and control.

[2] Zekri and Boullé, 2025. Fine-Tuning Discrete Diffusion Models with Policy Gradient Methods.

**Questions:**

* Length normalization is shown to have unintended consequences on the reasoning performance for AR LLMs. I am curious if it impacts the long reasoning behaviors in your experiments?

---

> ### Author Response · Authors · 2025-11-21
> **Response to Reviewer Twof**
>
> Thank you for your acknowledgment and constructive feedback on our paper. Below, we address your concerns and suggestions.
>
> ### **W1: how ELBO tightness affects policy improvement or bias across tasks?**
>
> In principle, the bias can indeed influence policy updates but limited. The bias of ELBO is a well-known characteristic in the dLLMs: prior work has discussed this gap on likelihood-based evaluation (e.g., perplexity[1*]) and training (e.g., DPO variants[2*]) and found that the ELBO is empirically a tight and practically effective surrogate for the exact likelihood. Since the exact likelihood is intractable for dLLMs, it is impossible to compare against an exact-likelihood policy objective directly. We have added this discussion to our revised manuscript in Section 2.1.
>
> ### **W2: Missed prior work baseline.**
>
> We agree that these baselines are important for a thorough empirical comparison. A discussion of method [3*] is added in Section 6 of the revised manuscript, and [4*] was already discussed in our original submission. We are currently implementing methods [3*] and [4*] on LLaDA models. We will report the results as soon as they are available and include the complete comparison in the camera-ready version.
>
>
>
> ### **Q: The impact of length normalization on long reasoning behaviors**
>
> In our experiments, we did not observe similar unintended consequences on long reasoning performance. We believe this is primarily due to our experimental setup, where the maximum generation length was fixed at 256 tokens.
>
> We found that our model's average output length was already near this limit at the start of RL training and consistently remained in the 220–256 token range throughout the process. This stable generation length, which is also consistent with observations in [5*]. Scaling the maximum generation length or variable-length generation methods (like block diffusion) may reveal different behaviors, which is left for future work.
>
>
>
> [1*] Aaron Lou, Chenlin Meng, and Stefano Ermon, 2023. Discrete diffusion modeling by estimating the ratios of the data distribution.
>
> [2*] Zhu et al, 2025. LLaDA 1.5: Variance-Reduced Preference Optimization for Large Language Diffusion Models.
>
> [3*] Venkatraman et al., 2024. Amortizing intractable inference in diffusion models for vision, language, and control.
>
> [4*] Zekri and Boullé, 2025. Fine-Tuning Discrete Diffusion Models with Policy Gradient Methods.
>
> [5*] Siyan Zhao et al, 2025. d1: Scaling reasoning in diffusion large language models via reinforcement learning.

---

### Author Response · Authors · 2025-11-23
**Summary of Paper Revision**

We thank all reviewers for their constructive feedback and have responded to each reviewer individually. We have also uploaded a **paper revision**, including additional results and illustrations. **All changes in revision are marked in blue.**


**For Reviewer Twof:**
- Section 2.1 (in Lines 104-107): We added a discussion on ELBO tightness and its validity as a surrogate for the exact likelihood.
- Section 6 (in Lines 462-463): We added a discussion on related work.

**For Reviewer 8CJA:**
- Appendix B (in Lines 729-755): We added detailed theoretical analysis on why the proposed sequence-level formulation is necessary and outperforms the token-level formulation.

**For Reviewer sYDr:**
- Appendix B (in Lines 729-755): We added detailed theoretical analysis on why the proposed sequence-level formulation is necessary and outperforms the token-level formulation.
-  Appendix C.2 and Figure 6 (in Lines 864-936): We included training curves for KL divergence and gradient norms using the $k_1$, $k_2$, and $k_3$ estimators, along with a detailed analysis.

**For Reviewer b9P2:**
- Appendix G.2 and Figure 8 (in Lines 1076-1116): We included the training curves for MATH and GSM8K tasks, along with a detailed analysis.
- Table 1 and Table 2 (in Lines 270-296): We added the missing baseline results.
- Appendix G.1 and Figure 7 (in Lines 1065-1103):  We added a detailed analysis of the sudden change observed in the Sudoku task, including additional training metrics such as gradient norm and KL divergence.
-  Appendix G.3 and Table 5 (in Lines 1123-1142):  We added additional ablations with d1 method using $k_2$ estimator.

---

### Author Response · Authors · 2025-12-03
**Summary of  Reviewer Consensus and Discussions Status**

Dear Area Chairs,

First, we sincerely appreciate your additional efforts in this unprecedented situation. To assist your evaluation, we provide a high-level summary of reviewer consensus and how we have addressed the reviewers’ concerns through clarifications and new experiments.
### **1. Core Contribution and Reviewer Consensus**
Our paper demonstrates that **sequence-level reinforcement learning (ESPO) is better aligned with the modeling characteristics of diffusion LLMs than token-level approaches**, achieving state-of-the-art results across mathematical reasoning, coding, and planning tasks. This contribution was recognized and highlighted by multiple reviewers., who highlighted three key strengths:

*   **Principled Solution for dLLMs:** ESPO effectively addresses the intractable token-level likelihood issue in diffusion LLMs by utilizing a sequence-level ELBO proxy (**Reviewer Twof, b9P2**).
*   **Simple yet Effective Formulation:** Reviewers also noted the formulation is simple to implement while avoiding heuristic surrogates (**Reviewer Twof,8CJA**).
*   **Strong Empirical Performance:** The method delivers consistent improvements over baselines (diffu-GRPO/d1, wd1) across diverse tasks (**Reviewer Twof, 8CJA, sYDr, b9P2**).

### **2. Summary of Discussions Status**
We would like to highlight the status of discussions prior to the system rollback:

**Group 1: Reviewers who have completed the discussion phase**
*   **Reviewer b9P2 (Initial: 4 $\to$ Post-Rebuttal: 6):**
    *   After we provided the requested missing baselines (Dream-7B + wd1), coding task benchmarks, and training stability analysis, the reviewer explicitly commented: *"Thank you for your answer, it solved my problem. I’ve increased rating to 6."*

**Group 2: Reviewers who have not yet responded to our rebuttal**
*   **Reviewer Twof (Initial: 8):**
    *  Initial review was highly positive. We have addressed their minor queries regarding ELBO tightness and added suggested missing citations in the revision.
*   **Reviewer 8CJA (Initial: 6):**
    *    Main concern was novelty compared to GSPO. We addressed this by adding a theoretical analysis (Appendix B) proving why sequence-level optimization is uniquely *necessary* for dLLMs (due to intractable token likelihoods ,which AR models do not face).
*   **Reviewer sYDr (Initial: 4):**
    *  Primary concern was theoretical grounding ("first principles"). In response, we formalized the derivation in the new Appendix B, showing that sequence-level optimization is computationally tractable while exact trajectory-level optimization is not, which helps contextualize ESPO as a practical and theoretically reasonable approximation. We also clarified the critical role of KL regularization with new experiments.

We hope that our additional clarifications and experiments have addressed the key concerns raised in the reviews. We also hope that this summary assists you in reviewing our submission more comfortably and efficiently.


Thank you again for your time and consideration.

Best regards,

The Authors

---

### Meta-Review · Area_Chair_ARCj · 2025-12-16

**Summary:**

The paper proposes ESPO, a sequence-level RL framework for diffusion LLMs that uses the ELBO as a tractable proxy for sequence likelihood and a k2 KL estimator for stability. Across two dLLMs (LLaDA-8B-Instruct, Dream-7B-Instruct) and multiple tasks (GSM8K/MATH, HumanEval/MBPP, Countdown/Sudoku), ESPO consistently outperforms token-level baselines (diffu-GRPO/d1, wd1), with especially large gains on planning (e.g., Table 1: +20–40 on Countdown; 60+ on Sudoku). The rebuttal added: (i) missing Dream+wd1 baselines; (ii) coding baselines against d1/wd1 (Table 2); (iii) training curves and KL/grad-norm diagnostics (Appendix C.2, G.1, G.2); (iv) an ablation showing d1+k2 remains weak vs ESPO (Appendix G.3 / Table 5); and (v) a theoretical appendix (Appendix B) justifying why sequence-level optimization with ELBO is the tractable choice for dLLMs vs token-level or full trajectory objectives. These additions resolve the empirical completeness concerns in R1 and the KL/implementation questions in R2, while partially addressing the novelty/first-principles concerns in R2/R3. Two issues remain: (1) no experimental comparison to Venkatraman et al. (2024) and Zekri & Boullé (2025), which the authors only discussed and deferred; (2) the theory stops short of a first-principles policy-gradient derivation for diffusion LMs (R2). Given the breadth of positive empirical evidence, principled formulation relative to existing token-level dLLM RL, and careful ablations, I recommend Accept (poster). The rebuttal largely overcame methodological/empirical objections; foundational questions about the exact policy-gradient form for dLLMs remain open but are not fatal for a poster-level contribution.

**Reviewer Concerns:**

#### Reviewer_b9P2
1. **Concern**: Missing training curves for math (MATH, GSM8K).
   - **Why Unresolved**: Resolved by rebuttal: Appendix G.2 and Figure 8 added curves and analysis.
   - **Impact on Decision**: None after resolution.

2. **Concern**: Missing Dream-7B + wd1 baseline.
   - **Why Unresolved**: Resolved by rebuttal: Dream-7B + wd1 added; ESPO > wd1 across tasks (Section 5/Table 1; author reply).
   - **Impact on Decision**: None after resolution.

3. **Concern**: Coding benchmarks lacked baseline comparisons.
   - **Why Unresolved**: Resolved by rebuttal: Table 2 now compares ESPO vs d1 and wd1 on HumanEval/HumanEval+ and MBPP/MBPP+; ESPO leads.
   - **Impact on Decision**: None after resolution.

4. **Concern**: Sudoku shows sudden performance jump; needs analysis.
   - **Why Unresolved**: Resolved by rebuttal: Appendix G.1/Figure 7 relate the jump to a spike in gradient norm and rapid KL rise; plausible learning transition.
   - **Impact on Decision**: None after resolution.

5. **Concern**: Is the gain mainly from the k2 KL estimator? Compare wd1/d1 with k2.
   - **Why Unresolved**: Resolved by rebuttal: wd1 has no explicit KL; d1+k2 ablation (Appendix G.3/Table 5) remains far below ESPO, indicating improvements are not solely due to k2.
   - **Impact on Decision**: None after resolution.

---

#### Reviewer_sYDr
1. **Concern**: Incremental novelty; lacks first-principles policy-gradient derivation tailored to diffusion LMs.
   - **Why Unresolved**: Partially addressed: Appendix B argues trajectory-level RL for dLLMs is intractable and motivates sequence-level ELBO as the only feasible surrogate, but it does not derive the exact diffusion-LM policy gradient from first principles nor quantify estimator bias comprehensively.
   - **Impact on Decision**: Moderate: limits theoretical claims and keeps the paper at poster level despite strong empirical results.

2. **Concern**: Hyperparameter selection fairness (PPO epsilon).
   - **Why Unresolved**: Resolved: authors used the same ε=0.2 across methods following prior work (author rebuttal).
   - **Impact on Decision**: None after resolution.

3. **Concern**: Role and form of KL regularization (why critical here, which estimator used).
   - **Why Unresolved**: Resolved with new diagnostics (Appendix C.2/Figure 6; Figure 2): k2 is stable; k1 collapses; k3 stagnates/spikes. Authors justify why sequence-level ELBO ratios amplify numerical issues, making KL crucial.
   - **Impact on Decision**: None after resolution.

4. **Concern**: Bias/variance rationale for sequence vs token formulation.
   - **Why Unresolved**: Partially addressed: Appendix B provides feasibility and modeling-alignment arguments; no quantitative bias-variance study. Empirical ablations (Figure 1) and d1+k2 (Table 5) support the claim.
   - **Impact on Decision**: Low-to-moderate: lack of quantitative bias-variance decomposition is a weakness but mitigated by strong empirical evidence.

---

#### Reviewer_8CJA
1. **Concern**: Novelty relative to GSPO; is this a straightforward application to diffusion LLMs?
   - **Why Unresolved**: Partially addressed: Appendix B explains token-level likelihoods are intractable for dLLMs and motivates ELBO-based sequence-level ratios; ESPO also integrates stabilized ELBO length-normalized ratios and k2 KL. Still, the algorithmic differences vs GSPO are not fundamentally new beyond adapting to the diffusion setting.
   - **Impact on Decision**: Moderate: reduces the paper from potential spotlight to poster.

2. **Concern**: Why per-token evaluation is problematic specifically for dLLMs (vs AR LMs).
   - **Why Unresolved**: Resolved: The rebuttal and Appendix B detail that token-level ELBO components lack a valid conditional-likelihood interpretation for dLLMs; empirical ablations (Figure 1) show token-level + ELBO is unstable.
   - **Impact on Decision**: None after resolution.

---

#### Reviewer_Twof
1. **Concern**: ELBO tightness and its impact on policy improvement/bias.
   - **Why Unresolved**: Partially addressed: Section 2.1 and discussion cite prior evidence of ELBO tightness; no task-level quantitative bias analysis. Empirically strong results lessen the concern.
   - **Impact on Decision**: Low: acceptable for poster, but precludes stronger recommendation.

2. **Concern**: Missing comparisons to Venkatraman et al. (2024) and Zekri & Boullé (2025).
   - **Why Unresolved**: Unresolved experimentally: discussed in Section 6, but authors deferred implementation to camera-ready; no numbers provided.
   - **Impact on Decision**: Moderate: holds back from a higher recommendation.

3. **Concern**: Effect of length normalization on long reasoning behaviors.
   - **Why Unresolved**: Resolved: authors report no adverse effects under fixed length 256; outputs remain 220–256 tokens; future work for larger/variable lengths.
   - **Impact on Decision**: None after resolution.
``

**Reviewer Scores:**

#### Reviewer_b9P2
- **Original Score**: 4
- **Expected Score After Discussion**: 6
- **Rationale**: The authors added the requested Dream+wd1 baseline, coding baselines (Table 2), math training curves, Sudoku diagnostics, and the d1+k2 ablation (Table 5). Reviewer explicitly raised to 6 post-rebuttal.

---

#### Reviewer_sYDr
- **Original Score**: 4
- **Expected Score After Discussion**: 4 or 6
- **Rationale**: The foundational ‘first-principles’ concern remains only partially addressed, but the authors provided a substantive Appendix B, strong KL analysis (Appendix C.2), and comprehensive empirical ablations; a modest increase is reasonable.

---

#### Reviewer_8CJA
- **Original Score**: 6
- **Expected Score After Discussion**: 6 or 8
- **Rationale**: Appendix B clarifies why sequence-level optimization is particularly necessary for dLLMs; added ablations and stronger baselines bolster empirical support. Novelty vs GSPO still limits the ceiling but merits a bump. 7 is a suitable score, but there is no 7

---

#### Reviewer_Twof
- **Original Score**: 8
- **Expected Score After Discussion**: 8
- **Rationale**: Their concerns were minor (ELBO tightness discussion, related work, length normalization). With the revisions, the strong-positive stance is likely maintained.

---

### Decision · Program_Chairs · 2026-01-26

Accept (Poster)